# Intermetallic PdZn nanoparticles catalyze the continuous-flow hydrogenation of alkynols to *cis*-enols

Xiao Chen [1,4], Chuang Shi[1,4], Xing-Bao Wang [2,3,4], Wen-Ying Li[2,3 ✉] & Changhai Liang [1 ✉]

Designing highly active and stable lead-free palladium-based catalysts without introducing surfactants and stabilizers is vital for large-scale and high-efficiency manufacturing of *cis*-enols via continuous-flow semi-hydrogenation of alkynols. Herein, we report an intermetallic PdZn/ZnO catalyst, designed by using the coupling strategy of strong electrostatic adsorption and reactive metal-support interaction, which can be used as a credible alternative to the commercial PdAg/Al$_2$O$_3$ and Lindlar catalysts. Intermetallic PdZn nanoparticles with electron-poor active sites on a Pd/ZnO catalyst significantly boost the thermodynamic selectivity with respect to the mechanistic selectivity and therefore enhance the selectivity towards *cis*-enols. Based on in situ diffuse reflectance infrared Fourier-transform spectra as well as simulations, we identify that the preferential adsorption of alkynol over enol on PdZn nanoparticles suppresses the over-hydrogenation of enols. These results suggest the application of fine surface engineering technology in oxide-supported metal (particles) could tune the ensemble and ligand effects of metallic active sites and achieve directional hydrogenation in fine chemical synthesis.

[1] State Key Laboratory of Fine Chemicals, Laboratory of Advanced Materials and Catalytic Engineering, School of Chemical Engineering, Dalian University of Technology, Dalian 116024, China. [2] State Key Laboratory of Clean and Efficient Coal Utilization, Taiyuan University of Technology, Taiyuan 030024, China. [3] Key Laboratory of Coal Science and Technology (Taiyuan University of Technology), Ministry of Education, Taiyuan 030024, China. [4]These authors contributed equally: Xiao Chen, Chuang Shi, Xing-Bao Wang. ✉email: ying@tyut.edu.cn; changhai@dlut.edu.cn

The semi-hydrogenation of alkynol to corresponding *cis*-enol is a critical process in the industrial production of fine and intermediate chemicals[1–3]. Typically, *cis*-2-butene-1,4-diol (*cis*-BED) derived from the hydrogenation of 2-butyne-1,4-diol (BYD) with the global demand of ca. 25,000 tons per annum can be used as an intermediate of endosulfan, vitamin A and B6, and as monomer to generate unsaturated telechelic polyether diol[3,4]. 2-methyl-3-buten-2-ol is an intermediate for industrial synthesis of vitamin E (ca. 140,000 tons per annum)[3]. Owing to the large market demand for enols, continuous-flow semi-hydrogenation of alkynols over heterogeneous catalysts will be an inevitable trend to obtain high-quality products under large-scale applications in virtue of their process safety, environmental friendliness, and economic benefits[5–10]. Generally, for achieving highly selective hydrogenation of alkynol to *cis*-enol, it is necessary to utilize the strategy of surface engineering for designing catalysts to enhance the thermodynamic selectivity with respect to mechanistic selectivity[11,12]. In the same instant, the hydrogenolysis or hydrodeoxygenation of alkynols should be inhibited for preventing the cleavage of C–OH bonds during the hydrogenation processing. In industry, the classic Lindlar catalyst is commonly utilized in the semi-hydrogenation of alkynol, in which the surface active sites blocked or isolated by the second metal Pb and reversible organic adsorbents boosted the selectivity to enol. Nevertheless, the introduction of heavy metal Pb and additives not only leads to the serious reduction of activity, but also causes the environment pollution[13]. Therefore, how to design Pb-free catalysts to avoid the selectivity–activity antinomy represents a severe challenge.

Intermetallic compounds (IMCs) with ordered crystal structures and special electronic structure play a key role considering the practical manufacturing of heterogeneous catalysts[14–17]. The selective hydrogenation of alkynols using Pd-based IMCs catalysts has brought about widespread attention[18–23]. The introduction of a second metal tuned the ensemble and ligand effects of metallic Pd significantly, enhancing the selective hydrogenation of alkynes[24–26]. Generally, supported IMCs can be designed by doping a second metal to promote or modify supported metal or reactive metal-support interaction under extremely harsh conditions[27]. Since Tauster et al. reported the strong metal-support interaction (SMSI) effect in 1978 first, tremendous efforts had been made to explore the structural reconstruction and electronic changes in the formation process of IMCs due to the SMSI effect and their influences on the catalytic performances[28–30]. However, the classic route based on the reactive metal-support interaction was prone to cause agglomeration and sintering of metal particles at high reduction temperatures due to the Ostwald ripening effect and facilitate the partial coverage of active centers at the metal surface by the reduced oxide support. Therefore, to achieve highly efficient support IMCs catalysts, it was certainly worth designing an oxide-supported metal processor that highly dispersed metal nanoparticles (NPs) with firm anchorage on the support.

Recently, the strong electrostatic adsorption (SEA) had been proved to be an effective strategy to anchor metal precursors on oxide supports and therefore achieved ultrasmall as well as homogeneous metallic NPs, in which precursors with a monolayer coverage were locally stabilized *via* the strong electrostatic interactions instead of the weak capillary forces induced by surface tension[31]. In this work, the relative coupling strategy of SEA and following high-temperature reduction was developed to design intermetallic PdZn/ZnO catalysts for continuous-flow semi-hydrogenation of alkynols. The hydrogen spillover invoked the partial reduction of ZnO and provoked the formation and growth of intermetallic PdZn NPs with electron-poor (Pd$^{\delta+}$) active sites. ZnH$_x$ species have been first detected to escape under

the high reduction temperature. Due to the SMSI effect as well as the strong anchoring effect of intermetallic PdZn NPs on the surface of ZnO, the optimized PdZn/ZnO catalyst exhibited superior activity, unique selectivity, as well as outstanding stability in the continuous-flow semi-hydrogenation of alkynols to *cis*-enols, which can be used as a credible alternative catalyst to the commercial PdAg/Al$_2$O$_3$ and Lindlar catalyst. This work explores different avenues to optimize the Pd-based catalysts for semi-hydrogenation of alkynols.

## Results and discussion

**PdZn/ZnO catalyst prepared by SEA and following reduction.** Site modification and isolation were known to be a powerful strategy of enhancing the selectivity of Pd-based catalysts for the semi-hydrogenation of alkynols to enols[32]. Here, a PdZn/ZnO catalyst has been designed by the relative coupling strategy of SEA and following high-temperature reduction, in which Pd active sties were modified by Zn *via* formation of intermetallic PdZn, as shown in Fig. 1a[33]. Due to the strong covalent Pd–Zn interaction, the pronounced charge correlation regions around the Pd atom owing to electron accumulation can be observed in Fig. 1b. The calculated results showed that the Mulliken atomic charges are ca. −0.17 and 0.18 e for the superficial Pd and Zn atom, respectively. Based on the fact that the *d*-bandwidth of metal directly affects its capacity to adsorb unsaturated chemical bonds by the semi-empirical method[34], the intermetallic PdZn/ZnO catalyst may present high potential in the selective hydrogenation of alkynols.

Generally, the chemical environment, especial for the pH of solution, has a tremendous effect on the metal ion adsorbed on metal oxide support. Supplementary Fig. 1 showed that the point of zero charge of ZnO (specific surface area = 18 m$^2$ g$^{-1}$) is ca. 8.2 and Na$_2$PdCl$_4$ as the Pd precursor could be achieved the maximum adsorption density under the optimal pH$_{Final}$ of ca. 5.9–6.3. As a consequence, high dispersion can be achieved in Pd/ZnO catalyst under the effect of high ionic strength. For exploring the SMSI effect on the Pd NPs and ZnO support, the Pd/ZnO precursor prepared by SEA with the theoretic metal loading of 1 wt% has been reduced from 150 to 500 °C under H$_2$/Ar atmospheres. As shown in Supplementary Fig. 2, X-ray diffraction (XRD) patterns of ZnO and Pd/ZnO-*T* samples (*T* referred to the reduction temperatures) only presented the typical diffraction peaks of ZnO, indicating that the well-dispersed Pd NPs on the ZnO support were too small to be detected. Supplementary Fig. 3 showed the N$_2$ adsorption–desorption isotherms of samples. The corresponding specific surface area, pore volume, and pore diameter were summarized in Supplementary Table 1. With strengthening the SMSI effect on the Pd NPs and ZnO, the specific surface area decreased while the pore volume increased slightly, which may be attributed to the structural reorganization of Pd/ZnO sample deriving from intermetallic PdZn NPs formed on the surface of ZnO.

Thermogravimetric/derivative thermogravimetric-mass spectrometry (TG/DTG-MS) measurement of Pd/ZnO sample in 5% H$_2$/Ar up to 520 °C was performed to survey the reduction process. As shown in Fig. 1c, the total mass loss of sample after the reductive treatment is 8.2 wt%, which was well correlated with the oxygen content, meaning that the material including the support was reduced under these harsh conditions. There were several reduction steps judging by the interrupted phase formation. (a) The reduction of PdO into Pd was prone to carry out below 150 °C. (b) As a result of hydrogen spillover from Pd NPs to ZnO, the partial ZnO support was reduced into Zn under 150–350 °C. In the same instant, the active Zn atoms doped into the space lattice of Pd NPs to form intermetallic PdZn. (c) There was obvious mass loss exceeding 350 °C, which could be

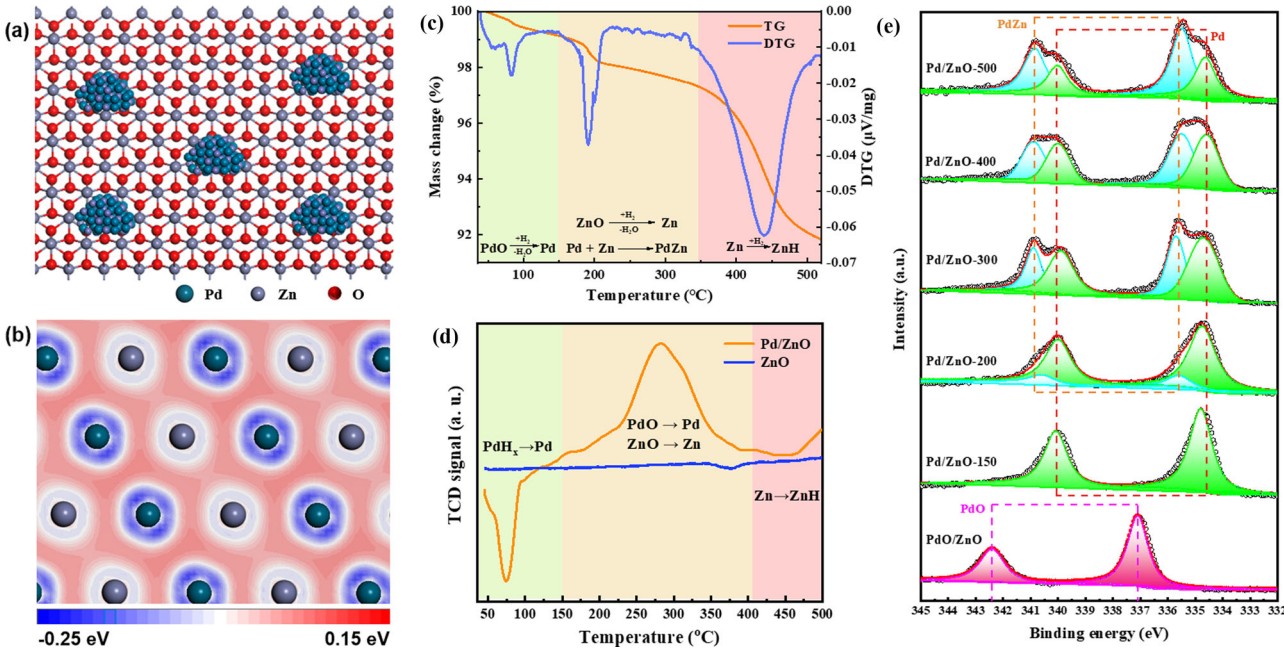

**Fig. 1 Structure characterization of intermetallic PdZn. a** Schematic illustrating PdZn/ZnO catalyst. **b** Deformation density of intermetallic PdZn. **c** TG/DTG profiles for the Pd/ZnO sample under 5 vol% $H_2$/Ar from room temperature to 520 °C with 5 °C $min^{-1}$. **d** $H_2$-TPR profiles of Pd/ZnO and ZnO samples in 10 vol% $H_2$/Ar up to 500 °C with 5 °C $min^{-1}$. **e** XPS spectra of Pd $3d$ region of unreduced PdO/ZnO and Pd/ZnO-$T$ samples.

attributed to the formation of $ZnH_x$ due to the hydrogen spillover effect as well as its sublimation[35,36]. The corresponding $m/z$ signal of 18 ascribed to water and $m/z$ signal of 66 ascribed to $ZnH_x$ of the exhaust were detected, as shown in Supplementary Fig. 4. Supplementary Fig. 5 displayed the weight ratio of Zn/Pd and the weight loss of Zn in the Pd/ZnO-$T$ samples detected by inductively coupled plasma atomic emission spectroscopy (ICP-AES), clearly showing that the sublimation of Zn after the reduction of sample at >450 °C. Temperature programmed reduction ($H_2$-TPR) is a powerful tool to detect the reducibility of Pd/ZnO and understand the SMSI between Pd NPs and ZnO support (as shown in Fig. 1d). Interestingly, a negative signal at ca. 72 °C was observed on the $H_2$-TPR profile of Pd/ZnO sample, which indicated that hydrogen releases directly because that Pd is apt to hydride-forming at ambient temperature. Noticeably, there was a broad reduction peak ca. 200–350 °C. Combined with the results of TG/DTG-MS, it can be conducted that PdO was reduced completely and then hydrogen spillover from Pd to the interface of Pd/ZnO, which led to a part of ZnO being reduced to produce lower valence $Zn^{(2−\delta)+}$ ($0 < \delta < 2$). In addition, there was no reduction peak in the $H_2$-TPR profile of pure ZnO, further confirming the self-catalysis of Pd/ZnO and the hydrogen spillover effect. Interestingly, the $H_2$-TPR signal of Pd/ZnO sample was staying upon the baseline at high temperature above 400 °C, which can be ascribed to the combination of reduced Zn with $H_2$ to produce $ZnH_x$. Ultimately, in the reduction process, intermetallic PdZn NPs are formed through two-step processes including the reduction of PdO NPs as well as ZnO support partially and subsequently the diffusion of Zn atoms into Pd lattices due to the SMSI effect, which will be confirmed by subsequent X-ray photoelectron spectroscopy (XPS) and high-angle annular dark-field scanning transmission electron microscopy (HAADF-STEM) measurements.

Figure 1e illustrates the XPS spectra of unreduced PdO/ZnO and Pd/ZnO-$T$ samples. There were two peaks at 342.4 and 337.1 eV for the unreduced PdO/ZnO sample, which can be ascribed to the Pd $3d_{3/2}$ and Pd $3d_{5/2}$ of PdO, respectively. The peaks assigned to PdO diminish while new peaks due to metallic

$Pd^0$ grow at 334.8 and 340.1 eV when the Pd/ZnO powder is reduced at 150 °C. With further increasing the reduction temperature, new peaks at 335.4 ± 0.1 eV and 340.7 ± 0.1 eV appear along with peaks for metallic $Pd^0$, which can be originated from the intermetallic PdZn phase formation[18,37]. It was well-known that the Pd $3d_{5/2}$ binding energy of $Pd^0$ has a positive shift (ca. 0.6 eV) in the presence of ZnO, which was compatible with the formation mechanism of PdZn phase. Pd/ZnO-$T$ ($T \geq 200$ °C) sample exhibited a higher shifts indicative of electron-poor ($Pd^{\delta+}$) sites, deriving from the electron transfer from Pd to ZnO support. The result was similar to theoretical calculation. Through calculating the molar ratio of PdZn/Pd based on the XPS peak area, it can be observed that the proportion of PdZn/Pd changes from 0.25 to 2.0 with the reduction temperature increasing from 200 to 500 °C (Supplementary Fig. 6). It is noticeable that multiple chemical states of metallic Pd and intermetallic PdZn coexisted on ZnO surface, which could be attributed to the sublimation of Zn in the $H_2$ atmosphere. It corresponded to the previous TG/DTG-MS and $H_2$-TPR measurements of PdO/ZnO sample. In addition, there was no unequivocal conclusion about the XPS spectra of Zn $2p_{3/2}$ region (Supplementary Fig. 7) due to the strong signal from ZnO support. However, a small shift with ca. 0.2 eV to lower binding energies was observed with increasing reduction temperature, which may be originated from the formation of Zn–OH groups on the Pd/ZnO-$T$ surface. Based on the above analysis, the SMSI effect was observed in the SEA-prepared Pd/ZnO system following direct reduction in $H_2$ ($T \geq 200$ °C), in which the electronic modification of Pd NPs by a SMSI effect and the intermetallic PdZn phase formation may play a central role in the semi-hydrogenation of alkynols *via* tuning the absorption/activation of substrates.

**Morphology of PdZn/ZnO catalyst.** The morphology and topography of the as-prepared Pd/ZnO-400 sample were revealed by TEM (Fig. 2a). It was suggested that particles with average size 7.0 nm were well-dispersed with a narrow size distribution (Fig. 2b). The typical HAADF-STEM images shown in Fig. 2c, d

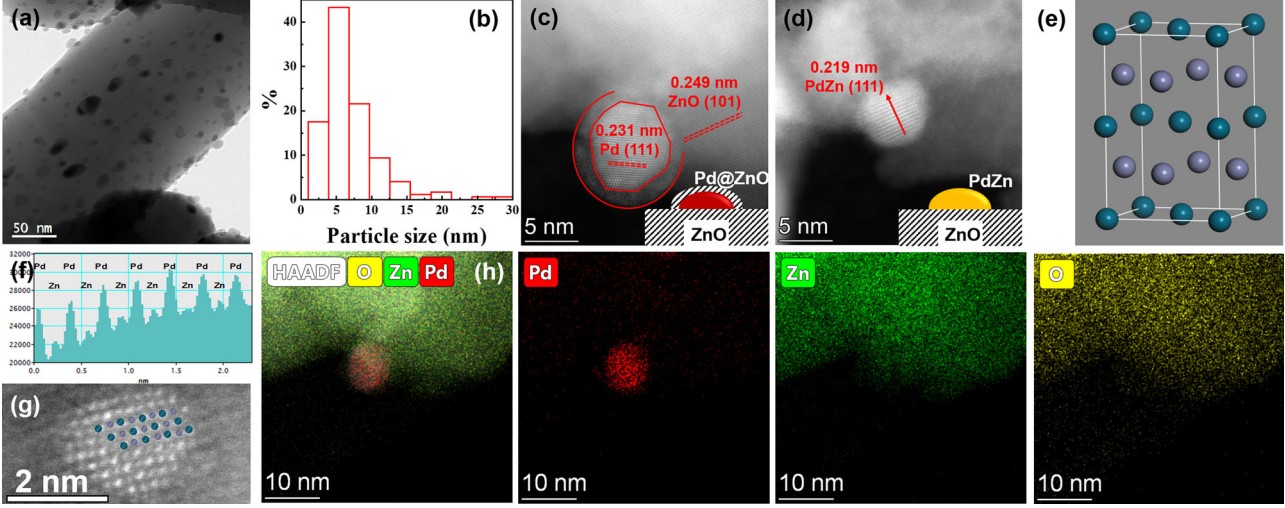

**Fig. 2 Morphologies of sample. a** TEM image, **b** the particle size distribution, and **c, d** HAADF-STEM images with insetting the corresponding models of Pd/ZnO-400 sample. **e** An atomic model of a unit cell of PdZn (Pd: dark cyan and Zn: indigo blue). **f** Intensity profile along the arrow in the HAADF-STEM image in **d**. **g** The higher magnification HAADF-STEM image of PdZn particles, displaying characteristic ordering of Pd and Zn atoms. **h** A HAADF-STEM image of Pd/ZnO-400 sample along with the corresponding EDS elemental maps of Pd, Zn, and O.

reveal the presence of both Pd and PdZn ordered surface atomic structure in the sample, which was confirmed by XPS results. The monometallic Pd NPs was featured by the atomic plane distance of 0.230 nm, matching with the lattice spacing of Pd (111). Interestingly, the Pd NPs with particle size 5.0 nm surrounded by ZnO overlayer was observed (Fig. 2c). Due to the metal-support interactions, Zn atoms migrate outside the Pd-rich NPs to form a stack on their surface. Essentially, the lattice spacing is measured as 0.219 nm for the particle in Fig. 2d, which was assigned to the (111) facets exposed on continuous PdZn nanocrystals with tetragonal structure (PdZn JCPDS 065-9523)[38,39]. The corresponding atomic model of a unit cell of PdZn (Pd: dark cyan and Zn: indigo blue) was shown in Fig. 2e. Figure 2f showed that the alternate distribution of Pd and Zn elements was further substantiated by the intensity profile along the arrow in Fig. 2d. High-resolution HAADF-STEM images of representative PdZn particles for the Pd/ZnO-400 (Fig. 2g) showed that the projection of the theoretical atomic structure of PdZn IMCs superimposes on the experimental image, which was characterized by alternating smaller Zn and bigger Pd atoms. The compositions of intermetallic PdZn/ZnO were further confirmed by HAADF-STEM image and X-ray energy dispersive spectroscopy (EDS) mapping. It clearly showed both Pd and Zn signals of particle at the edge of the ZnO support (Fig. 2h).

**Selective hydrogenation of BYD.** Generally, the hydrogenation of BYD involves parallel and consecutive isomerization as well as hydrogenation reactions, generating several key intermediates, such as cis-BED, trans-BED, butane-1,4-diol (BDO), butanol (BOL), 4-hydroxybutanal, and tetrahydrofuran-2-ol (2-OH-THF), as shown in Fig. 3a. The catalytic performance of SEA-prepared Pd/ZnO-T catalysts in 5 wt% BYD aqueous system was evaluated at 2 MPa $H_2$, 80 °C, and at contact time of 29 $g_{cat.}$·h·mol$^{-1}$. As shown in Fig. 3b, the conversion of BYD was reduced at first and then increased with the increasing reduction temperature. Under the reactive metal-support interaction between Pd NPs and ZnO, the active centers of the metal surface were partially blocked by a thin layer of ZnO support. However, at high reduction temperatures, the reconstruction of catalyst as well as the formation of intermetallic PdZn continued to occupy center stage, which enhanced the ensemble and ligand effects on the Pd active sites, boosting the semi-

hydrogenation of BYD to cis-BED. It clearly showed that the selectivity to cis-BED can reach ca. 92% with the BYD conversion of 96% over the as-prepared Pd/ZnO-400 catalyst. The intrinsic catalytic activities reflected by TOFs of Pd/ZnO-T catalysts are shown in Supplementary Table 1. The TOF value of Pd/ZnO-400 catalyst with the PdZn/Pd molar ratio 1:1 (71 min$^{-1}$) was slightly lower than that of Pd/ZnO-150 catalyst with pure Pd NPs supported on ZnO (90 min$^{-1}$), which further indicated that partial active sites were converged. Under the same reaction conditions, 100% conversion of BYD can be achieved over monometallic Pd-based catalysts, but the product distributions are relatively complicated. The selectivity to cis-BED was only 7.5% for Pd/Al$_2$O$_3$, 1.5% for Pd/SiO$_2$, 0.4% for Pd/C, 19.3% for Pd/TiO$_2$, and 94.3% for Pd/CaCO$_3$, respectively. The over-hydrogenation of cis-BED was prone to carry out over monometallic Pd-based catalysts, especially for the Pd/SiO$_2$ and Pd/C catalysts with weak metal-support interaction. In addition, the hydrogen-mediated cis-trans BED isomerization was observed on Pd/TiO$_2$ catalyst and the selectivity to trans-BED reaches 28.9%. This phenomenon was similar to selective stereochemical catalysis over RhSb/SiO$_2$[40]. Therefore, Zn isolated Pd active centers could tune the adsorption of substrates and intermediates and inhibit the side reactions. This result was similar to previous phenomenon that intermetallic PdZn with Pd–Zn-Pd ensembles for semi-hydrogenation of acetylene[41]. By comparison with the commercial PdAg/Al$_2$O$_3$ catalyst and the industrial Lindlar-type catalyst, the Pd/ZnO-400 catalyst contained PdZn as active sites presented promising application prospects in which the partial doping of second metals as well as the SMSI effect boost the selectivity to cis-BED but without introducing toxic additives.

Supplementary Fig. 8 and Fig. 3c, d showed the typical product distributions and the selectivity of the hydrogenation of BYD over Pd/ZnO-150 and Pd/ZnO-400 catalysts, respectively. The initial reaction rate of Pd/ZnO-400 catalyst is ca. 12.66 μmol s$^{-1}$ g$_{cat.}$$^{-1}$, which was higher than that over Pd/ZnO-150 catalyst (9.45 μmol s$^{-1}$ g$_{cat.}$$^{-1}$). Significantly, the selectivity to cis-BED over Pd/ZnO-400 catalysts maintained a relatively stable production with higher than 90.0% as contact time increased (Fig. 3d). The SMSI effect with respect to Pd–Zn covalent interaction partially blocked the non-selective active sites of Pd NPs preferred the thermodynamic selectivity comparing to the mechanistic selectivity, thus suppressing the over-hydrogenation of cis-BED to the product of BDO. This was similar observation on the energetically favorable

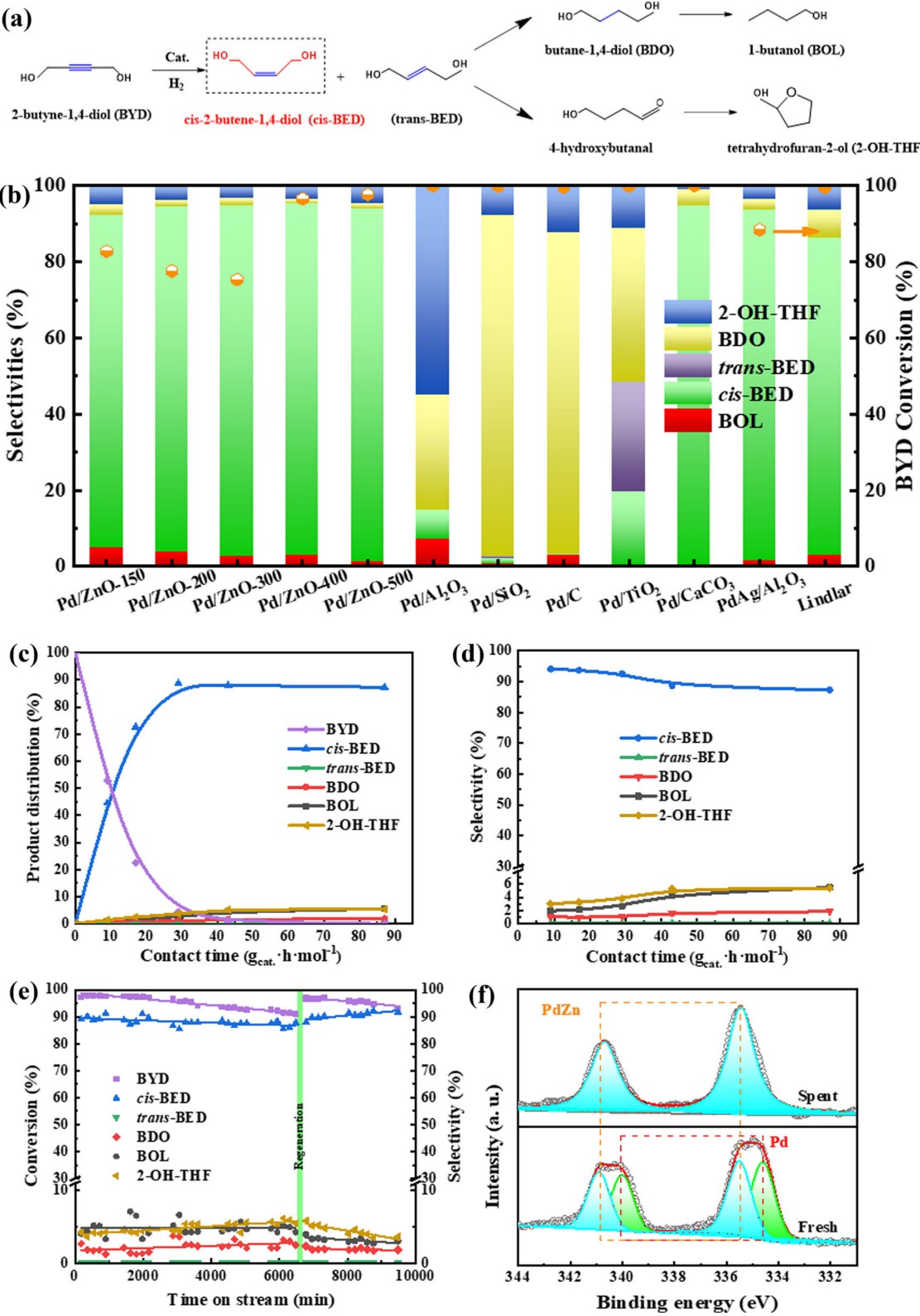

**Fig. 3 Selective hydrogenation properties of catalysts. a** Reaction pathway for the hydrogenation of BYD. **b** Conversion of BYD and product selectivity over Pd/ZnO-*T* catalysts and the commercial Pd-based catalysts (Reaction conditions: 2 MPa H$_2$, 80 °C, and contact time 29 g$_{cat}$ h mol$^{-1}$). **c, d** Product distribution and selectivity of the BYD hydrogenation versus the contact time over Pd/ZnO-400 catalyst. **e** The stability testing of Pd/ZnO-400 catalyst in the BYD hydrogenation, in which the catalyst is regenerated at 110 h. **f** XPS spectra of Pd 3*d* region of the fresh and spent Pd/ZnO-400 catalyst.

path for acetylene hydrogenation and ethylene desorption over the ultrasmall intermetallic PdZn particles[42]. In addition, the formation of Zn–OH groups on the Pd/ZnO-$T$ surface (observed by XPS spectra in Supplementary Fig. 7) may inhibit the adsorption/activation of C–OH groups in BYD and BED, weakening the hydrogenolysis reaction, resulting the lower selectivity to by-products (BOL and 2-OH-THF) (lower than 4.0% over Pt/ZnO-400 catalyst). As a comparison, the hydrogenation of BYD over Pd/ZnO-400 catalyst was also evaluated in a batch reactor. As shown in Supplementary Fig. 9, the Pd/ZnO-400 catalyst also presented high efficiency in the selectivity hydrogenation of BYD to *cis*-BED. However, the products obtained from the reaction in the continuous-flow reactor had a higher stability at the fixed reaction conditions than in the batch reactor. In addition, it is usually adopted for industrial scale-up through the selectivity hydrogenation of BYD in the continuous-flow reactor.

The effects of reaction temperatures, $H_2$ pressures, and the $H_2$/substrate volume ratios on the catalytic performances in the selectivity hydrogenation of BYD over Pd/ZnO-400 catalyst in the continuous-flow reactor were also explored, and the results were shown in Supplementary Figs. 10–12. The selective hydrogenation of BYD to *cis*-BED was suitable under mild conditions. High selectivity to *cis*-BED (>97%) at almost complete conversion of BYD can be achieved at 80 °C, 0.5 MPa, and 100 $H_2$/substrate ratio. Although the conversion of BYD was boosted under hard conditions, it easily led to the formation of by-products and lowering the selectivity to intermediate *cis*-BED. Supplementary Fig. 10 showed that *cis*-BED was prone to isomerization and over-hydrogenation at high temperature, while the over-hydrogenation of C=C bond to C–C bond and the cleavage of C–OH bond were enhanced under high $H_2$ pressure (Supplementary Fig. 11). In addition, the selective hydrogenation of BYD to *cis*-BED in the continuous-flow reactor can be operated at the relatively low $H_2$/substrate volume ratio with 100 from the perspective of hydrogen economics (Supplementary Fig. 12) and it may be necessary to recycle hydrogen in the industrial scale-up.

**Stability testing**. The time course of BYD hydrogenation activity over Pd/ZnO-400 catalyst at 80 °C for 160 h under 2 MPa $H_2$ pressure with a contact time of 29 $g_{cat.}$·h·mol$^{-1}$ was evaluated. Figure 3e showed that the BYD conversion was initially ca. 98% as well as the selectivity to *cis*-BED was ca. 90% at 3 h. With the prolongation of the stability reaction time to 110 h, the conversion of BYD showed a slight decrease to ca. 90%. Excitingly, the selectivity to *cis*-BED presented a relative stability. In the same instant, a few by-products including BDO (sel.% = 2%), BOL (sel.% = 5%), and 2-OH-THF (sel.% = 4%) were detected. It can be deduced that the non-selective PdH$_x$ active sites with high activity in the surface of Pd/ZnO-400 catalyst were buried in oblivion as the extension of the reaction time. The reconfiguration of Pd/ZnO-400 surface with the adsorption of BYD molecules may be carried out. The corresponding oxide and hydroxide was easily formed on the surface of Pd or PdZn in aqueous phase hydrogenation environment for a long term. In situ regeneration by $H_2$ reduction treatment is a powerful strategy to eliminate the strongly adsorbed species on the active sites and strengthen the interaction between inter(metallic) active center with oxides support[43]. After terminating the liquid feedstock and in situ reducing the catalyst under 48 sccm hydrogen flow at 400 °C for 2 h, the second stage stability testing was carried out at the same reaction conditions. The initial conversion of BYD (98%) is regained with respect to ca. 90% selectivity to *cis*-BED. Up to 160 h, the Pd/ZnO-400 catalyst still showed outstanding stability and excellent *cis*-BED selectivity of >90%.

Through analyzing the content of metals in the reaction solution by ICP-AES, no leaching of Pd and Zn was observed (below the detection limit). The structural stability of spent Pd/ZnO catalyst was determined from measurements of phase state and carbon content by using XPS and TG/DTG, respectively. As shown in Fig. 3f, there are two peaks at 340.8 and 335.4 eV for the spent Pd/ZnO-400 sample, which can be ascribed to the Pd $3d_{3/2}$ and Pd $3d_{5/2}$ of PdZn intermetallic compound, respectively. Comparing with the fresh sample, the peaks assigned to metallic Pd$^0$ disappear, which may be attributed to the reconstruction of catalyst under the second reduction. The interaction between Pd NPs and ZnO support was strengthened and expressed by the formation of intermetallic PdZn. In addition, it was noteworthy that no obvious mass loss peak was observed in the high-temperature region for the TG/DTG curve of spent Pd/ZnO-400 catalyst (Supplementary Fig. 13), which meant that there was no carbon species formed on the catalyst during the hydrogenation of BYD in aqueous phase, probably deriving from that the isolation of Pd active sites by Zn in turn resists carbon deposition. The results reflected that intermetallic PdZn NPs supported on ZnO was a potential catalyst for the semi-hydrogenation of BYD due to its hydrothermal stability as well as coke resistance.

**Substrate scopes**. To demonstrate the general applicability of intermetallic PdZn catalyst, the semi-hydrogenation of other industrially relevant alkynols was also tested over the Pd/ZnO-400 catalyst, and the results were shown in Supplementary Fig. 14. As expected, this material exhibited a remarkable chemo-selective semi-hydrogenation of 3-hexyn-1-ol, 3-phenyl-2-propyn-1-ol, and 2-methyl-3-butyn-2-ol to their corresponding *cis*-enols, respectively. Significantly, the selectivity to these enols presented quite stably with increasing the contact time. It further confirmed that the adsorption of enols was rather weaker than that of alkynols on the surface of intermetallic PdZn NPs, which suppressed over-hydrogenation. As presented in Table 1, the TOFs for the hydrogenation of these alkynols over Pt/ZnO-400 catalyst was as follows: 2-methyl-3-butyn-2-ol (397 min$^{-1}$) > 3- hexyn-1-ol (157 min$^{-1}$) > 3-phenyl-2-propyn-1-ol (110 min$^{-1}$) > BYD (71 min$^{-1}$). It clearly showed that the steric-hindrance effect derived from the substrate molecule played a vital role in the hydrogenation of alkynols. Among these reactions, the hydrogenation of alkynol with terminal alkyne group was relatively easy to carry out. The above results indicated that the reactive metal-support interaction was an effective strategy to design highly efficient Pd-based catalysts for the semi-hydrogenation of alkynols to enols.

**In situ DRIFT spectroscopy of the hydrogenation of propargyl alcohol**. In situ diffuse reflectance, infrared Fourier-transform (DRIFT) spectroscopy was utilized to identify the spectroscopic properties of reactive surface intermediates in the selective hydrogenation of alkynol over Pd/ZnO catalyst. As shown in Fig. 4, the adsorption of propargyl alcohol at 30 °C as well as its hydrogenation on the surface of Pd/ZnO-400 catalyst with SMSI effect was investigated. Compared with the ATR infrared spectroscopy of pure propargyl alcohol (as shown in Fig. 4a), the time-resolved spectra of sample exposed under gas-phase propargyl alcohol and evacuated under Ar displayed the typical surface species. There was a peak at 3294 cm$^{-1}$, which was consistent with the stretching vibration of an alcohol hydroxyl group (ν(OH)). In the same instant, the vibrational at ca. 1030 cm$^{-1}$ was generally attributed to the stretching vibration of a single C–O bond in an alkoxy group[44]. However, the adsorption and reaction of alkyne led to a various of hydrocarbon fragments on surfaces[45]. Peaks at 2119[1] and 2035 cm$^{-1}$ could be attributed to the vibrational features for the acetylenic link in propargyl alcohol, while the peaks

**Table 1 The semi-hydrogenation of other industrially relevant alkynols over Pd/ZnO-400 catalyst.**

| Entry | Substrate | Product | Solvent | T (°C) | P (MPa) | $\tau$ [a] ($g_{cat.}\cdot h\cdot mol^{-1}$) | Conv. (%) | Sel. (%) | TOFs [b] ($min^{-1}$) |
|-------|-----------|---------|---------|--------|---------|----------|-----------|----------|-------|
| 1 | HO⟶OH | HO⟶OH | water | 80 | 2 | 29 | 95.8 | 92.6 | 71 |
| 2 | Ph⟶OH | Ph⟶OH | methanol | 100 | 3 | 67 | 88.5 | 90.0 | 110 |
| 3 | ⟶OH | ⟶OH | methanol | 130 | 3 | 41 | 97.7 | 97.2 | 157 |
| 4 | HO⟶ | HO⟶ | water | 50 | 1 | 29 | 98.5 | 95.8 | 397 |

[a] $\tau$ meant the contact time.
[b] The TOFs was calculated with the initial conversion of substrate <30%.

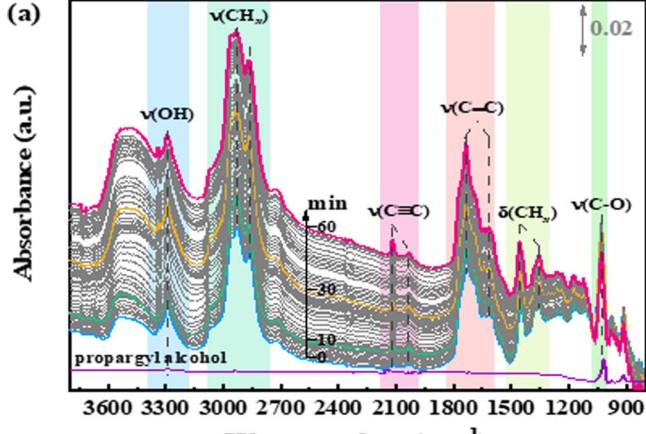

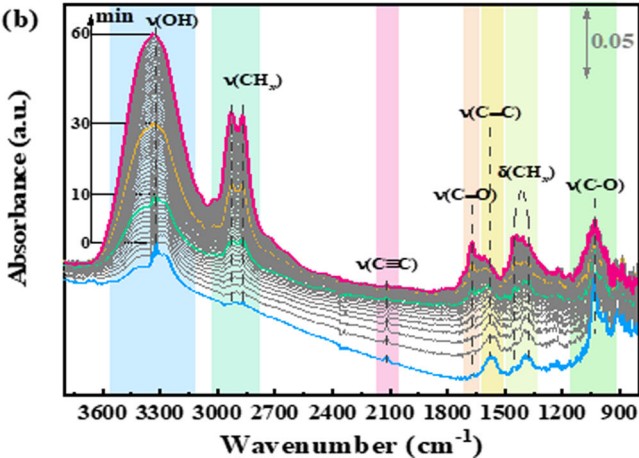

**Fig. 4 In situ DRIFT spectra.** DRIFT spectra of the reaction of propargyl alcohol over the Pd/ZnO-400 catalyst at 30 °C in **a** Ar and **b** 10 vol% H₂/Ar.

at 1737 and 1612 cm$^{-1}$ were assigned to π- and σ-bonded C=C fragments[46,47]. The surface absorbed H species may lead to the hydrogenation of the C≡C bonds to C=C bonds. In addition, the vibrational features for the CH and CH$_2$ functional groups were detected on Pd/ZnO-400 surface. According to reports that acetylene could be isomerized to vinylidene, stably existing on Pd. It clearly showed that peaks among 2800 and 3000 cm$^{-1}$ ascribed to CH stretches while peaks at 1442 and 1356 cm$^{-1}$ assigned to ethylidyne-type species as well as vinylidene-type species on the

Pd/ZnO-400 catalyst. As expected, propargyl alcohol strongly adsorbed on the Pd/ZnO-400 surface. The infrared spectroscopy was almost unchanged even that the adsorption species on the surfaces are evacuated under Ar for 60 min at 30 °C.

In addition, the reaction process of propargyl alcohol in the H$_2$/ Ar flow was monitored by infrared spectroscopy (as shown in Fig. 4b). Interestingly, the peaks at 2035 cm$^{-1}$ due to the vibrational features for the acetylenic link and 1737 cm$^{-1}$ assigned to π-bonded C=C fragment disappeared initially, which may be due to the greater susceptibility of these active species to hydrogen. This was an essential factor for the high hydrogenation activity of the Pd/ZnO-400 catalyst. Comparing with the time-resolved spectra of Pd/ZnO-150 sample (Supplementary Fig. 15), it can be discovered that the intensity of peak at 2119 cm$^{-1}$ identified to the adsorption of C≡C bonds weaken gradually while that of peak at 1574 cm$^{-1}$ classified to σ-bonded C=C fragment was almost unchanged with the increasing the reaction time, which indicated that the adsorbed propargyl alcohol was hydrogenated to propenyl alcohol rapidly over Pd/ZnO-400 catalyst. Similar with that the adsorption of C$_2$H$_2$ on Pd/ZnO catalyst was stronger than that of C$_2$H$_4$, the adsorption of propenyl alcohol was rather weaker than that of propargyl alcohol, so that it desorbed from the surface of catalyst quickly, suppressing its over-hydrogenation to produce n-propanol and leading to the high selectivity to intermediate. Meanwhile, the intensity of vibrational features for the CH$_x$ species on the surface of Pd/ZnO-400 catalyst increased with increasing the reaction time, which further confirmed the propargyl alcohol was hydrogenated. However, the intensity of peak at 1669 cm$^{-1}$ assigned to the C=O vibration enhanced gradually with prolonged reaction time more than 30 min, which may be attributed to the isomerization of enol[48]. Therefore, the key issue on highly selective hydrogenation of alkynol to enol was to improve the thermo-dynamic selectivity comparing to the mechanistic selectivity through designing the chemo-selective catalysts such as IMCs NPs.

**DFT analysis of reaction mechanism.** To better understand the underlying mechanism, the calculated total electronic density of states of BYD, BED, Pd, and PdZn and the adsorption config-urations and energies of reaction species over Pd (111) and PdZn (111) were calculated. As shown in Supplementary Fig. 16, the d bands center of intermetallic PdZn presented a negative deviation and the bandwidth was reduced significant in comparison with metallic Pd, which may be ascribed to the missing d–d interac-tions and lattice strain, leading to the dramatic change in the catalytic properties[49]. Interestingly, the d bands of PdZn over-lapped with the p bands of BYD slightly but there was almost no overlay with that of BED. However, they overlapped and

interweaved among the $d$ bands of metallic Pd with the $P$ bands of BYD and BED. From the point of electronic effect, it can deduce that intermetallic PdZn boosts the selectivity to BED and suppresses its over-hydrogenation comparing the metallic Pd due to the weaker interaction[21]. Based on the thermodynamically stable and mostly exposed surfaces of PdZn (111) and Pd (111), the configurations and energies of the most stable BYD, *cis*-BED, and *trans*-BED adsorption on these surfaces were calculated (Supplementary Fig. 17). The adsorption energies of BYD on the PdZn (111) and Pd (111) were −0.97 and −1.30 eV, respectively, emerging highly efficient reactivity of Pd sites of the PdZn (111) and Pd (111). Moreover, the adsorption energies of *cis*-BED on the PdZn (111) and Pd (111) were −0.27 and −0.55 eV, respectively, which represented that the desorption of *cis*-BED on the surface of PdZn had priority over that on Pd. The same results can also be found for the *trans*-BED while the *cis*-*trans* BED isomerization should be further explored. In view of the above-mentioned facts, it can deduce that the thermodynamic selectivity has been enhanced and the over-hydrogenation or hydrogenolysis of BED over intermetallic PdZn may be inhibited significantly. Regulating the Pd active sites to the completely isolated by Zn given rise to preferential π-adsorption of C≡C bonds of BYD, and decreased the adsorption energies significantly, indicating the intermetallic PdZn is a promising catalyst for the semi-hydrogenation of BYD to BED.

The energy distribution diagrams of the hydrogenation pathways of C≡C bonds in BYD were depicted in Fig. 5. Similar with the adsorption of 2-methyl-3-butyn-2-ol, the C≡C groups of BYD were more activated by Pd (111) and PdZn (111) comparing with that of C−OH groups[21]. Clearly, the BYD hydrogenation barriers over Pd (111) were 0.8 eV (TS1$_{Pd}$), 1.39 eV (*trans*-TS2$_{Pd}$), 1.11 eV (*cis*-TS2$_{Pd}$), 2.06 eV (TS3$_{Pd}$), and 0.87 eV (TS4$_{Pd}$), respectively (Fig. 5a). The first energy barrier for the *cis*-BED hydrogenation was in fact much higher than the one corresponding to the C≡C bonds hydrogenation in BYD, suggesting that the *cis*-BED (ad) hydrogenation over the Pd (111) was the rate-determining step. While the hydrogenation barriers of *trans*-TS2$_{Pd}$ was close to that of *cis*-TS2$_{Pd}$, indicating that cis/trans-isomerism of BED can be obtained over metallic Pd catalyst. In contrast, the hydrogenation barriers of BYD over the PdZn (111) were 0.62 eV (TS1$_{PdZn}$), 2.49 eV (*trans*-TS2$_{PdZn}$), 1.00 eV (*cis*-TS2$_{PdZn}$), 2.23 eV (TS3$_{PdZn}$), and 0.97 eV (TS4$_{PdZn}$), respectively (Fig. 5b). The higher energy of TS3$_{PdZn}$ implied that *cis*-BED preferred desorption rather than further hydrogenation over intermetallic PdZn. The result was consistent with the hydrogenation of acetylene over atomically ordered intermetallic PdZn NPs[41,42]. Interestingly, the value of *cis*-TS2$_{PdZn}$ was much lower than that of *trans*-TS2$_{PdZn}$, which indicated that there was extremely unlikely to carry out the cis/trans-isomerism of BED over intermetallic PdZn. Highly selective hydrogenation of BYD to *cis*-BED can be achieved over intermetallic PdZn due to the strengthen of thermodynamic selectivity derived from the isolated Pd active sites. This theoretical simulation result nicely interpreted the experimental results as shown in Fig. 3 and the reaction mechanism of the selective hydrogenation of BYD over intermetallic PdZn was proposed, as shown in Fig. 6. Drawing on the experience that the selective hydrogenation acetylene to intermediate ethylene could be evaluated by the barrier difference ($\Delta E_a$) between the hydrogenation and the desorption of ethylene[50]. Generally, an ideal catalyst ought to possess a more positive $\Delta E_a$ value. By comparing Fig. 5 with Supplementary Fig. 17, it can be clearly found that the $\Delta E_a$ of PdZn (111) and Pd (111) were calculated to be 1.96 and 1.51 eV, respectively, additional illustrating that intermetallic PdZn was more effective in obtaining high selectivity of *cis*-BED. The above DFT calculation further demonstrated that the atomic regulation of Pd sites by in situ

formation of intermetallic PdZn was a promising strategy to optimize alkynol semi-hydrogenation processes.

**Conclusion.** In summary, we demonstrated here that a facile and controllable reduction of SEA-prepared noble metal NPs supported on metal oxides provided a powerful strategy in tailoring their catalytic performance by boosting the SMSI effect as well as inducing the formation of IMCs. In Pd/ZnO system, hydrogen spillover invoked the partial reduction of ZnO and provoked the formation of electron-poor (Pd$^{\delta+}$) active sites due to the electron transfer from Pd to Zn. Compared with the commercial PdAg/Al$_2$O$_3$ catalyst and the traditional Lindlar catalyst, Pd/ZnO-400 catalyst exhibited outstanding performance in semi-hydrogenation of BYD, achieving higher than 90% selectivity to *cis*-BED at 100% conversion. In addition, long-term stability of Pd/ZnO-400 catalyst was attained at aqueous phase hydrogenation environment. Pd/ZnO-400 catalyst also showed broad scope activity and unique selectivity to enols toward semi-hydrogenation of other alkynols. Combining the in situ *DRIFTS* analysis and DFT theoretical calculations, we demonstrated that the thermodynamic selectivity comparing to the mechanistic selectivity had been enhanced for the hydrogenation of alkynols through isolating the Pd active sites and modifying their electronic structure. The relative coupling strategy of strong electrostatic adsorption and following high-temperature reduction will open a surface engineering route to promote and optimize the semi-hydrogenation of Pd-based catalysts.

## Methods

**Preparation of Pd/ZnO-*T* catalysts by SEA method.** The strategy of SEA was employed for achieving the high dispersion of Pd NPs on ZnO support. Briefly, the functional groups on the ZnO surface could be protonated or deprotonated by adjusting the pH value to form a strongly charged surface, thereby anchoring oppositely charged metal ions. The point of zero charge of ZnO support was detected first and Na$_2$PdCl$_4$ as the Pd precursor was chose to adsorbed under acidic conditions (see Supporting Information 2). To determine the pH at which the strongest surface interaction occurs, metal salt uptakes at various pH values were measured by ICP-AES, using a surface loading of 1000 m$^2$ L$^{-1}$. After electrostatic adsorption, the sample were filtered, dried at 100 °C overnight, and then reduced in H$_2$ with a steady flow of 20 sccm for 2 h at various temperatures (150, 200, 300, 400, and 500 °C) with a heating rate of 5 °C min$^{-1}$.

**Preparation of Pd/Al$_2$O$_3$ catalysts by wet impregnation.** 0.14 g Na(Pd)Cl$_4$ was dissolved in 30 mL H$_2$O after ultra-sonication for 30 min. Then 5.0 g Al$_2$O$_3$ was added into the solution and stirred for 2 h to acquire a homogeneous suspension. The Pd/Al$_2$O$_3$ sample was successfully synthesized after elimination of H$_2$O by vacuum distillation with a rotary evaporator at 50 °C, calcination at 400 °C for 2 h in 50 vol% O$_2$/Ar, and then reduction at 400 °C in 30 vol% H$_2$/Ar for 2 h.

**Characterization of catalysts.** The specific surface areas and pour structure of samples were analyzed by a Quantachrome Autosorb iQ automated gas sorption analyzer. XRD analyses of the samples were carried out using a Rigaku D/Max-RB diffractometer equipped with a Cu Kα$_1$ monochromatized radiation source. The practical loading of Pd in the samples and the content of Pd and Zn in the reaction solution were analyzed by ICP-AES. TG/DTG analysis of the PdO/ZnO sample and the spent Pd/ZnO-400 catalyst was performed on a Mettler-Toledo TGA/SDTA3+ jointing with Pfeiffer OmniStar mass spectrometer. Both H$_2$-TPR and chemisorption of CO were carried out on AutoChem II 2920 Chemisorption Analyzer with a thermal conductivity detector. It was supposed that the adsorption stoichiometric ratio is Pd:CO = 1:1 for the whole samples. STEM images and elemental mapping were obtained adopting the aberration-corrected microscope operating at 200 kV. The XPS measurements were performed on Thermo VG Scientific ESCALAB250 electron spectrometers using a monochromatic Al Ka X-ray source. All Pd/ZnO-*T* samples were pre-reduced at different temperatures before test and then transferred into the chamber by the Schlenk technology while the spent Pd/ZnO-400 catalyst was directly moved into the chamber for the characterization.

**In situ DRIFTS spectroscopy.** Thermo Nicolet i50 FTIR spectrometer with a Praying Mantis DRIFTS reactor (Harrick Scientific) was used to detect the adsorption or hydrogenation of propargyl alcohol on Pd/ZnO sample. The spectra were collected at a resolution of 4.0 cm$^{-1}$ for 16 scans in the region of 400–4000 cm$^{-1}$. Before collecting the IR spectrum, ca. 0.1 g Pd/ZnO sample loaded on the DRIFTS reactor

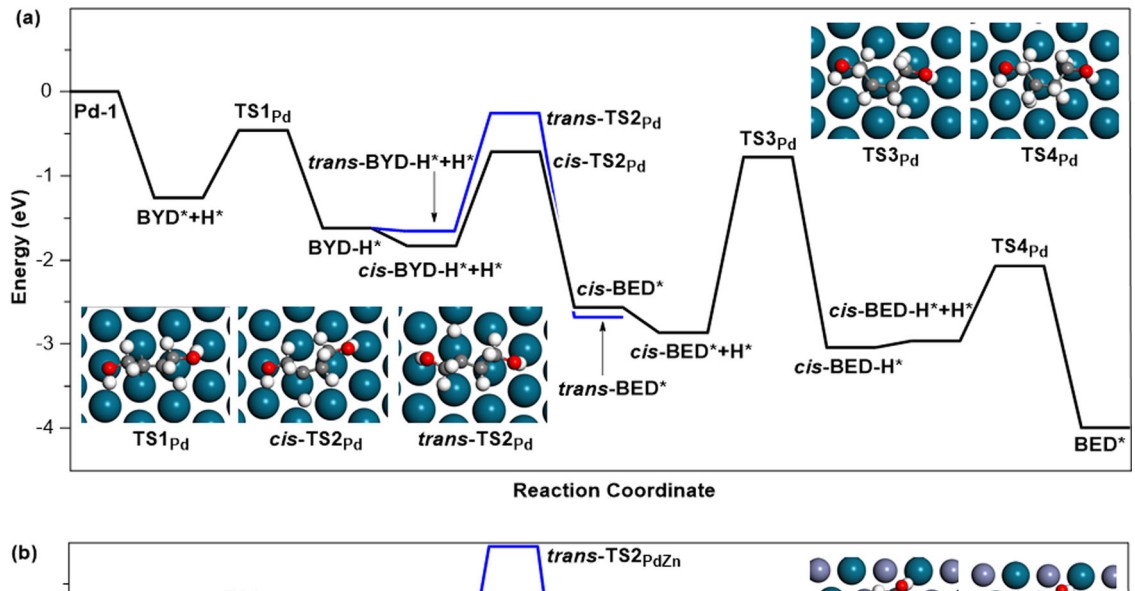

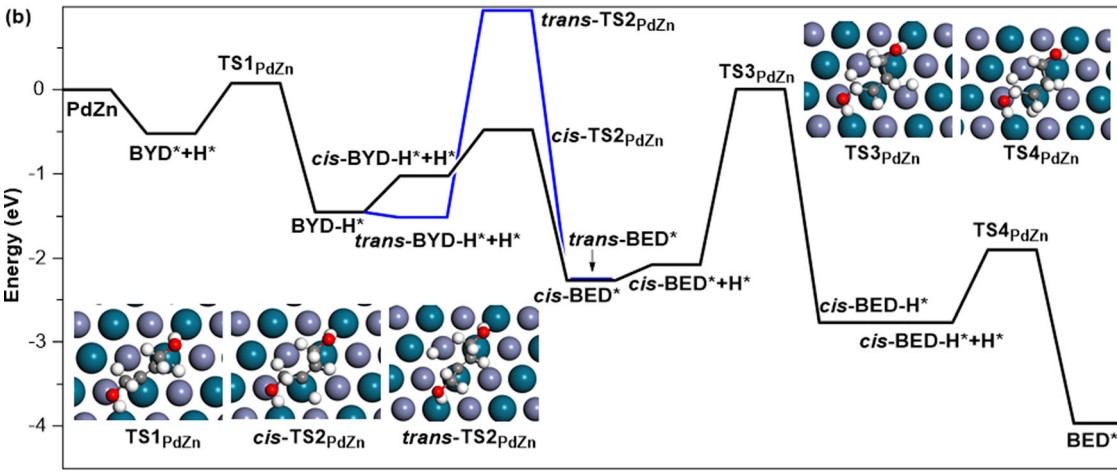

**Fig. 5 Transition states of BYD hydrogenation.** Potential energy diagram by theoretical calculation for the hydrogenation of C≡C bonds in BYD over **a** Pd (111) and **b** PdZn (111).

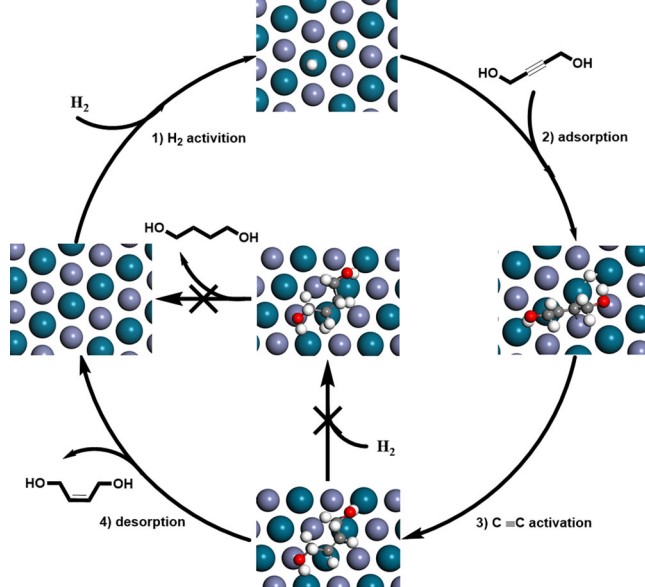

**Fig. 6 Reaction mechanism.** Reaction mechanism of the selective hydrogenation of BYD over intermetallic PdZn.

was in situ reduced in 5 vol% $H_2$/Ar for 2 h with the flow rate of 40 sccm at 150 and 400 °C, respectively, and then cooled down to 30 °C in Ar, at which point a background spectrum was acquired. Subsequently, propargyl alcohol under the action of Ar as carrier gas was fed into the system until the adsorption was saturated. After removing any physisorbed and gas-phase propargyl alcohol through purging with pure Ar, the spectra of intermediate products on the surface reaction in the $H_2$ flow (40 sccm) were captured.

**Computational**. The Pd (111) and PdZn (111) surfaces were modeled on p(6 × 6) supercells involving four-layered slabs with the vacuum layer height of 15 Å, in which the top layer was permitted to relax for optimizing the adsorption configuration. All calculations were performed using the Doml[3] package. The generalized gradient approximation (GGA) with the Perdew–Burke–Ernzerhof function (PBE) was selected in this study. The calculation parameters were similar to those used in the previous research (see Supporting Information 3)[51].

**Liquid-phase hydrogenation of alkynols**. The liquid-phase selective hydrogenation of alkynols over Pd/ZnO-*T* samples, Pd/Al₂O₃, and the commercial Pd-based catalysts (Pd/C, Pd/TiO₂, Pd/SiO₂, Pd/CaCO₃, PdAg/Al₂O₃, Lindlar) were performed in a continuous-flow reactor system consists of a stainless steel pipe (length, 750 mm; internal diameter, 8 mm) (Supplementary Scheme 1). For exploring the SMSI effect on the catalytic performance, 0.1 g as-prepared Pd/ZnO sample diluted with quartz to 5 mL was in situ activated with $H_2$ (48 sccm) for 2 h at various temperature (150, 200, 300, 400, and 500 °C) before testing. Typically, the liquid reactant composed of 5 wt% BYD aqueous solution and 1 wt% 1,2-propylene glycol (as internal standard) was pumped into the reactor to attend the hydrogenation reaction. The effects of reaction temperatures (room temperature-100 °C), $H_2$ pressures (0.1–2.0 MPa), the volume ratio of $H_2$-to-liquid feed (50–600), and the contact time (0–87 $g_{cat.}$ h mol⁻¹) on the catalytic performances

in the selectivity hydrogenation of BYD over Pd/ZnO-400 catalyst were explored. By comparison, the BYD hydrogenation performance over Pd/ZnO-400 catalyst in the batch reactor was also carried out (see Supporting Information 4). For exploring the applicability of Pd/ZnO catalyst in the semi-hydrogenation of alkynols, the hydrogenation of 3-hexyn-1-ol, 3-phenyl-2-propyn-1-ol, and 2-methyl-3-butyne-2-ol has also been tested in fixed bed reactor over Pt/ZnO-400 catalyst, respectively. The products were analyzed by gas chromatography (Agilent 7890B-GC) equipped with a flame ionization detector. The carbon balance of products was within ±5% with no gas generated. The contact time, the conversion (C%) of alkynol, and the selectivity ($S_i$%) of products were defined as follows:

$$\text{Contact time}(g_{cat.} \, h \, mol^{-1}) = m_{cat.}/(F*60) \quad (1)$$

$$C\% = (n_0 - n_t)/n_0 \times 100\% \quad (2)$$

$$S_i\% = n_i/\Sigma n_i \times 100\% \quad (3)$$

where $m_{cat.}$ (g) was the mass of catalyst (0.1 g), F (mol min$^{-1}$) was the molar flow rate of reactant, $n_0$ and $n_t$ assigned to the mole of reactant in the feed and product, respectively, $n_i$ attributed to the mole of product $i$ and $\Sigma n_i$ are the total moles of products.

For understanding the intrinsic catalytic activity of Pd/ZnO-$T$ catalysts, the turnover frequency (TOF) was evaluated by the follow formula:

$$TOF(min^{-1}) = (F \times C\%)/(n_{total} \times D) \quad (4)$$

where C% was the conversion of reactant, $n_{total}$ was the molar of metal in catalyst, and D was the metal dispersion that calculated from the CO chemisorption.

## Data availability
The data that support the findings of this study are available from the corresponding author upon reasonable request.

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

## Acknowledgements

We acknowledge with pleasure the support of this work by the Science and Technology Innovation Fund in Dalian City (2019J12GX028 and 2018RQ09), and the Project Partenariats Hubert Curien (PHC) Cai Yuanpei grant.

## Author contributions

X.C. conceived the idea and designed the experiments. C.S. conducted the experiments. X.-B.W. conducted theoretical calculation and simulation. X.C. and C.S. wrote the paper, and all authors discussed the results and comment on the manuscript. W.-Y.L. and C.L. were the project supervisors. We thank Dr. Chi-Wing Tsang's contribution for improving English expression.

## Competing interests

The authors declare no competing interests.
