## [Peer Review File · Communications Chemistry]

Reviewers' comments:

Reviewer #1 (Remarks to the Author):

This manuscript describes the design and preparation of highly active and stable Pb-free Pd-based catalysts for continuous-flow semi-hydrogenation of alkynols. Catalytic tests show that the optimized PdZn/ZnO-400 catalyst exhibits superior activity, unique selectivity, as well as outstanding stability in the continuous-flow semi-hydrogenation of alkynols to cis-enols, which can be as a credible alternative catalyst to the commercial PdAg/Al₂O₃ and Lindlar catalyst. These results are interesting, therefore I recommend to publish this work after minor revisions in the following:

1. As shown in Figure 2, authors should show details for XPS measurements. Were the tests performed at in-situ conditions or not?
2. Authors should show more details for the reaction mechanism, and I suggest to draw a proposed scheme.
3. Authors should give the Pd particle size distribution. Are there any smaller Pd particle size?

Reviewer #2 (Remarks to the Author):

The authors describes the synthesis of intermetallic PdZn/ZnO catalysts for the semihydrogenation of alkynols in flow reactor. The article doesn't offer strong element of novelty both in term of materials and catalysis, already described in literature and not sufficiently cited. Although the article offer some interesting perspectives in my opinion it can't be considered suitable for the publication for several reasons, cited below. The manuscript is unacceptable in its present form. The study seem promising and the authors should be encouraged to consider resubmission in the future, probably in other journal with several review.

- 1) The manuscript is difficult to read and confusing. The lack of schemes of reaction, tables o schemes in the text, an overuse of acronyms (BID, BED, SMSI, RMSI IMC, SEA, DOS etc), an excessive reference to schemes in SI, make it hard to read . I suggest you to modify these points and to improve the english.
 - 2) At pag 5, the author introduce the catalyst PdZn/ZnO-400 but he doesn't explain the meaning of "400". Only after we can deduce it's the "reduction temperature".
 - 3) The synthesis of such catalyst was explained in part at the beginning (pag 7-8, with reference to Figure S1 in SI, and then again at the end (pag 25) with other details. In my opinion It's not so clear. Please explain better all details for the synthesis at the beginning or at the end. .
 - 4) The article offer an accurate characterization of catalysts but it's extremely lacking in term of data of catalysis (e.g. experiments of catalysis, parameters to evaluate the productivity of catalysts, an accurate description of the flow reactor, comparison with commercial catalysts).
- The authors use a flow reactor but they doesn't describe the instrument (Volume, length, materials, connections; check controls, process schemes) and the advantages respect to a batch

reactor neither in the text nor in SI. They didn't report comparisons in batch conditions, please insert.

- Tables with complete data of conversions, selectivity, STY, TOF, TON, productivity, should be added for hydrogenation of 2-butyne-1,4-diol but also for 2-methyl-3-butyne-2-ol, 3-hexynol, 3-phenyl-2-propyne-1-ol, described at pag 18 only with TOFs.

- The authors report only the contact time but it's not clear if they change only the flow of substrate, maintaining the content of catalyst (0.100 g ???) or both. In a flow process It's very important to show how productivity changes on the basis of parameters such as Hydrogen-flow, substrate-flow, time-contact, etc. Please specify.

- TOFs values obtained are good and in line with literature data but using hard reaction conditions in term of Pressure (2MPa), temperature (80°C), H₂/substrate ratio=600.

- I suggest you to add experiments at different temperatures (e.g. room temperature), low pressures (e.g. 1-2 bar) and low H₂/Substrate ratio, changing the hydrogen flow. This type of reactions doesn't need hard conditions. Mild conditions favour selectivity.

- Authors compare their catalyst only with Lindlar catalyst and Pd/ Al₂O₃. Please add comparison with other commercial catalysts (e.g. Pd/C; Pd/TiO₂; Pd/SiO₂; Pd/CaCO₃) in the same reaction conditions

- Please describe better in situ regeneration by H₂, reaction conditions and characterization of catalyst fresh, used and regenerated.

- No information about leaching in solution was considered. Please measure metal leaching in solution by ICP.

5) the article doesn't offer strong element of novelty both in term of materials and catalysis, already described in literature and not sufficiently cited (e.g. among them: for materials: ACS Catal 2017,7,2, 1491; Catal. Today 2018, 330, 2; J. Phys. Chem C 2011,115, 8457; ACS Appl. Nanomaterials 2019, 2,5,3307; for Hydrogenation of alkynes: Chem Eng Proc. Proc Intensification 2020, 154, 108018; ACS Sust Chem Eng 2019, 7, 13, 11050; J. Catal, 2014, 311, 212; Beilstein. J. Org Chem 2017, 13, 734; ChemCatChem 2017, 9, 3245). Please add some references for both arguments.

6) The suppression of overhydrogenation of BED to BOD, and the selectivity to cis-BED, cited at pag 16, can be explained by DFT analysis at pag 22 ?? Explain better. It's not clear. In case, please add in the text, at pag 16, some reference to this part .

7) Why do you make DRIFT spectroscopy with propargyl alcohol ? Why do you use such a dangerous/toxic compound, never used in previous cited catalytic experiments ? Why don't you use 2-butyne-1,4-diol or others?

8) DFT calculations are reported at the beginning (pagg 5-7) and then at the end of the paper (pagg

22-23) with reference to figures and schemes at the beginning and to SI. It make difficult to read and to understand the reasoning. I suggest to write only a chapter about DFT calculation at the end of the paper to explain all details for thermodynamics and mechanistic selectivity. In general, for a better understanding the text, I suggest to follow a scheme: synthesis of catalyst, complete characterization of catalyst, catalytic experiments in flow and batch, DFT calculations and DRIFT spectroscopy to explain the selectivity and mechanism, conclusion, methods. Now everything it's mixed and confusing.

Reviewer #3 (Remarks to the Author):

The manuscript entitled 'Hydrogen induced intermetallic PdZn for invoking the continuous-flow hydrogenation of alkynols to cis-enols' (COMMSCHEM-21-0260-T) by Chen et al. presents results from an experimental study on the preparation, characterisation and evaluation of new intermetallic PdZn catalysts for use in the selective semi-hydrogenation of 2-butyne-1,4-diol. The authors describe a targeted study of this new catalyst for the use in a special semi-hydrogenation where high selectivity towards the partially reduced alkene intermediate is desired and has significant industrial relevance. The industry standard for this reaction are Pb-containing noble metal catalysts such as Lindlar type catalysts, which have issues associated with their Pb content. In comparison the Pb-free catalyst the authors present here, even show improved selectivity compared to Lindlar catalysts. The authors' characterisation of this new PdZn catalyst is highly detailed and includes TG, X-ray spectroscopy methods such as XPS and EDX, TEM, DRIFT spectroscopy, and a DFT analysis of the reaction mechanism.

The manuscript is structured logically and written in most parts in clear English, however in certain sections it needs to be rewritten substantially to meet the standard of Commschem. In these sections, the language, expressions or grammar used are inadequate and negatively impact comprehensibility, up to the point where certain statements are rendered unintelligible; especially the section named 'Selective hydrogenation of alkynols', p16ff, requires significantly more work before it can be accepted. In general, the conclusions the authors have drawn are sound and the findings are novel, making this investigation an interesting read for a wider catalysis, organic chemistry and chemical engineering audience. Because the high demand for a highly selective, and ideally Pb-free, catalyst for these semi-hydrogenations, and the detailed characterisation of the new PdZn catalyst the authors have undertaken, I can recommend for this manuscript to be published in Commschem, after some changes outlined below and rewriting of the sections flagged above.

This is a list of changes that should be addressed before resubmitting a revised version:

1) Page 6, line 19: "Moreover, the adsorption energies of cis-BED on the PdZn (111) and Pd (111) are -0.27 eV and -0.55 eV, respectively..." – what about the adsorption energies of trans-BED? In the experimental hydrogenation section the authors have found that the alkene predominantly produced is cis not trans, which agrees with most literature. Nevertheless, information on the adsorption energies of trans-BED might be useful here to complete the picture.

2) Page 8, line 11: The term ZnO-T should be introduced here. This is done later on Page 26:"The obtained catalysts were assigned as Pd/ZnO-T, where T referred to the reduction temperatures. " - this should be done here.

3) Page 9, figure 2: The diagram 2d is not clear. How do the differently coloured images of the catalyst correlate to the graph? The link needs to be explained.

4) Page 10, line 19: "Ultimately, in the reduction process, intermetallic PdZn NPs are formed through two-step processes including the reduction of PdO NPs as well as ZnO support partially and subsequently the diffusion of Zn atoms into Pd lattices", What proof is there for migration of Zn into the Pd lattice?

5) Page 12, figure 3: It is not clear from the marking on the TEM images how distance measurements were conducted. What's the double dashed line on a), b) and c)?

6) Page 13, line 20: "The catalytic performance of SEA-prepared Pd/ZnO-T catalysts in 5 wt.% BYD aqueous system was evaluated at 2 MPa H₂, 80 °C, and contact time 29 gcat·h·mol⁻¹." How was the contact time derived?

7) Page 15, figure 4: "(a) Conversion of BYD and product selectivity over Pd-based catalysts (Reaction conditions: 2 MPa H₂, 80 °C, and contact time 29 gcat. h mol⁻¹)." – The bar graphs in figure 4a) show performance of catalysts prepared under different reduction temperatures; this is not made clear in the caption and also higher up in the text (end of page 13) and therefore the description should be altered to avoid confusion.

8) Page 16ff: "Significantly, the selectivity to cis-BED over Pd/ZnO-400 catalysts maintains a relatively stabilization with higher than 90.0% with increasing contact time (Figure 4(c)). The SMSI effect with respect to Pd-Zn covalent interaction partially blocked the non-selective active sites of Pd NPs enhances the adsorption of the BYD with respect to that of the cis-BED, thus suppressing the over-hydrogenation of BED to product of BDO.", this section and following sections need to be rewritten to make it clear what the authors would like to say.

9) Page 17, line 11: "This is the first report about the long-term stability of Pd-based catalyst in a continuous-flow semi-hydrogenation of BYD to cis-BED." This is not the first continuous flow study of Pd-based catalysts for the semi-hydrogenation of BYD to cis-BED! Very substantial work has been carried out over the past years on the semi-hydrogenation of this particular molecule, 2-butyne-1,4-diol. The authors should reference recent work on this reaction, especially in industry relevant continuous flow hydrogenation settings, such as the work from Moreno-Marrodan et al (Beilstein J Org Chem. 2017; 13: 734–754, doi: 10.3762/bjoc.13.73), Tanielyan et al (Org. Process Res. Dev. 2017, 21, 3, 327–335, <https://doi.org/10.1021/acs.oprd.6b00375>) and Kundra et al (Chem. Eng. Process. 2020, 154, 108018, <https://doi.org/10.1016/j.cep.2020.108018>).

10) Page 17, line 22: "weightlessness peak" – What's a weightlessness peak, can the authors explain?

11) Page 20, section named "In situ DRIFT spectroscopy of the hydrogenation of propargyl alcohol" – this section is very lengthy and parts of it should be moved to the supporting information.

12) Page 28, line 8: "...and H₂-to-liquid feed ratio of 600." – Is this feed ratio based on volume/volume or mass/mass?

C H Hornung

Reviewer #1:

1. As shown in Figure 2, authors should show details for XPS measurements.

Were the tests performed at in-situ conditions or not?

Reply: Thank you very much for your comments. In our experiments, the XPS measurements were performed on Thermo VG Scientific ESCALAB250 electron spectrometers using a monochromatic Al Ka X-ray source. The C1s peak at 284.6 eV was used as standard for the calibration of binding energy. Due to the limitation of experimental conditions, the tests performed at non in-situ conditions. However, all Pd/ZnO-*T* samples were pre-reduced at different temperatures (150 °C, 200 °C, 300 °C, 400 °C, and 500 °C) before test and then transferred into the chamber by the Schlenk technology.

Modification: In section named 'Characterization of catalysts', the details for XPS measurements have been presented.

All Pd/ZnO-*T* samples were pre-reduced at different temperatures before test and then transferred into the chamber by the Schlenk technology while the spent Pd/ZnO-400 catalyst was directly moved into the chamber for the characterization.

2. Authors should show more details for the reaction mechanism, and I suggest to draw a proposed scheme.

Reply: Thank you very much for your suggestion. In our work, the mechanism of the selective hydrogenation of BYD to *cis*-BED has been investigated basing on the *in situ* DRIFT spectroscopy and DFT calculations. Due to the isolated Pd active sites and modification their electronic structure by doping the Zn atoms, the BYD

hydrogenation and *cis*-BED desorption are energetically favorable over the intermetallic PdZn particles. The over-hydrogenation and *cis-trans* isomeric reaction of BED over intermetallic PdZn have been extremely inhibited. Therefore, high selectivity to *cis*-BED with complete conversion of BYD has been achieved in the continuous-flow hydrogenation of BYD over intermetallic PdZn particles. As your suggestion, the reaction mechanism has been proposed, as shown in Scheme 1.

Modification:

This theoretical simulation result nicely interprets the experimental results as shown in Figure 3 and the reaction mechanism of the selective hydrogenation of BYD over intermetallic PdZn was proposed, as shown in Scheme 1.

Scheme 1 Reaction mechanism of the selective hydrogenation of BYD over intermetallic PdZn.

3. Authors should give the Pd particle size distribution. Are there any smaller Pd particle size?

Reply: Thank you very much for your suggestion. The Pd particle size distribution has been detected by TEM. As shown in Figure 2(a), it can be found that the particles with average size of 7.0 nm were well-dispersed on the ZnO support.

Modification:

The morphology and topography of the as-prepared Pd/ZnO-400 sample was revealed by TEM (Figure 2(a)). It is suggested that particles with average size 7.0 nm were well-dispersed with a narrow size distribution.

Figure 2. Morphologies of samples. (a) TEM image of Pd/ZnO-400 sample with inset showing the particle size distribution. (b-c) HAADF-STEM images of Pd/ZnO-400 sample with inset showing the corresponding model of catalyst and an atomic model of a unit cell of PdZn. (d) Intensity profile along the arrow in the HAADF-STEM image in (c) and the higher magnification HAADF-STEM image of PdZn particles, displaying characteristic ordering of Pd (navy blue) and Zn (sky blue) atoms. (e) A HAADF-STEM image of Pd/ZnO-400 sample along with the corresponding EDS elemental maps of Pd, Zn, and O.

Reviewer #2:

1) The manuscript is difficult to read and confusing. The lack of schemes of reaction, tables in the text, an overuse of acronyms (BID, BED, SMSI, RMSI IMC, SEA, DOS etc), an excessive reference to schemes in SI, make it hard to read. I suggest you to modify these points and to improve the English.

Reply: Thank you very much for your comments. For better understanding the selective hydrogenation of alkynols, the scheme of reaction has been moved from Scheme S1 into the Figure 3(a). The results of the semi-hydrogenation of other industrially relevant alkynols over Pd/ZnO-400 catalyst has been listed in Table 1, which has been supplied in the revised manuscript. In addition, some acronyms in the manuscript have been replaced by the full name, Such as (RMSI: reactive metal-support interaction; DOS: density of states). According to your suggestion, the expression in English has been improved with the assistance of Dr. Chi-Wing Tsang, Technological and Higher Education Institute of Hong Kong.

Modification:

- The scheme of reaction has been added in to Figure 3(a).

Figure 3. Selective hydrogenation properties of catalysts. (a) Reaction pathway for the hydrogenation of BYD. (b) Conversion of BYD and product selectivity over Pd/ZnO-*T* catalysts and the commercial Pd-based catalysts (Reaction conditions: 2 MPa H₂, 80 °C, and contact time 29 g_{cat.} h mol⁻¹). (c-d) Product distribution and selectivity of the BYD hydrogenation versus the contact time over Pd/ZnO-400 catalyst. (e) The stability testing of Pd/ZnO-400 catalyst in the BYD hydrogenation, in which the catalyst is regenerated at 110 h. (f) XPS spectra of Pd 3d region of the fresh and spent Pd/ZnO-400 catalyst.

Table 1 The semi-hydrogenation of other industrially relevant alkynols over Pd/ZnO-400 catalyst

Entry	Substrate	Product	Solvent	T (°C)	P (MPa)	τ^a (g _{cat.} ·h·mol ⁻¹)	Conv. (%)	Sel. (%)	TOFs ^b (min ⁻¹)
1			water	80	2	29	95.8	92.6	71
2			methanol	100	3	67	88.5	90.0	110
3			methanol	130	3	41	97.7	97.2	157
4			water	50	1	29	98.5	95.8	397

^a τ means the contact time.

^b The TOFs is calculated with the initial conversion of substrate <30%.

➤ The expression in English, such as:

The selective hydrogenation of alkynols using Pd-based IMCs catalysts have brought about widespread attention [18-23].

Therefore, to achieve highly efficient support IMCs catalysts, it was certainly worth designing an oxide-supported metal processor that highly dispersed metal nanoparticles (NPs) with firm anchorage on the support.

Additionally, there was no unequivocal conclusion about the XPS spectra of Zn 2p_{3/2} region (as shown in Figure S7) due to the strong signal from ZnO support.

Under the same reaction conditions, 100% conversion of BYD can be achieved over monometallic Pd-based catalysts, but the product distributions are relatively complicated.

Significantly, the selectivity to *cis*-BED over Pd/ZnO-400 catalysts maintained a relatively stable production with higher than 90.0% as contact time increased (Figure 3(d)).

Significantly, the selectivity to these enols presented quite stably with increasing the

contact time.

2) At page 5, the author introduces the catalyst PdZn/ZnO-400 but he doesn't explain the meaning of "400". Only after we can deduce it's the "reduction temperature".

Reply: Thank you very much for pointing it. The physical meaning of T should be assigned at first mention. In the revised manuscript, we have introduced it at page 8 and removed it in section of Methods.

Modification:

"the optimized PdZn/ZnO-400 catalyst" has been changed into "the optimized PdZn/ZnO catalyst".

As shown in Figure S2, XRD patterns of ZnO and Pd/ZnO- T samples (T referred to the reduction temperatures) only presented the typical diffraction peaks of ZnO, indicating that the well-dispersed Pd NPs on the ZnO support are too small to be detected.

3) The synthesis of such catalyst was explained in part at the beginning (page 7-8, with reference to Figure S1 in SI, and then again at the end (page 25) with other details. In my opinion It's not so clear. Please explain better all details for the synthesis at the beginning or at the end.

Reply: Thank you very much for your suggestion. For better understanding the synthesis of Pd/ZnO catalysts by SEA method, all synthesis process analysis has been presented at the beginning with reference to Figure S1 in Supporting Information, and the synthesis method has shown in the experimental section.

Modification:

➤ At the beginning

Generally, the chemical environment, especial for the pH of solution, has a tremendous effect on the metal ion adsorbed on metal oxide support. Figure S1 showed that the point of zero charge of ZnO (specific surface area = $18 \text{ m}^2 \text{ g}^{-1}$) is ca. 8.2 and Na_2PdCl_4 as the Pd precursor could be achieved the maximum adsorption density under the optimal pH_{Final} of ca. 5.9-6.3. As a consequence, high dispersion can be achieved in Pd/ZnO catalyst under the effect of high ionic strength.

➤ At the end

Preparation of Pd/ZnO-T catalysts by SEA method. The strategy of SEA was employed for achieving the high dispersion of Pd NPs on ZnO support. Briefly, the functional groups on the ZnO surface could be protonated or deprotonated by adjusting the pH value to form a strongly charged surface, thereby anchoring oppositely charged metal ions. The point of zero charge of ZnO support was detected firstly and Na_2PdCl_4 as the Pd precursor was chose to adsorbed under acidic conditions (as described in the Supporting Information (Section S2)). To determine the pH at which the strongest surface interaction occurs, metal salt uptakes at various pH values were measured by ICP-AES, using a surface loading of $1000 \text{ m}^2 \text{ L}^{-1}$. After electrostatic adsorption, the sample were filtered, dried at $100 \text{ }^\circ\text{C}$ overnight, and then reduced in H_2 with a steady flow of 20 sccm for 2 h at various temperatures ($150 \text{ }^\circ\text{C}$, $200 \text{ }^\circ\text{C}$, $300 \text{ }^\circ\text{C}$, $400 \text{ }^\circ\text{C}$, and $500 \text{ }^\circ\text{C}$) with a heating rate of $5 \text{ }^\circ\text{C min}^{-1}$.

4) The article offer an accurate characterization of catalysts but it's extremely lacking in term of data of catalysis (e.g. experiments of catalysis, parameters to

evaluate the productivity of catalysts, an accurate description of the flow reactor, comparison with commercial catalysts).

Reply: Thank you very much for your comments. The related information of the liquid-phase hydrogenation of alkynols has been supplied in the revised manuscript, including the instrument, the parameters (reaction temperature, H₂ pressure, contact time, the amount of catalyst, and the concentration of liquid reactant). Additionally, the hydrogenation of 2-butyne-1,4-diol over commercial catalysts as comparison has been described in the experimental section. The details are as the follow responses.

- **The authors use a flow reactor but they doesn't describe the instrument (Volume, length, materials, connections; check controls, process schemes) and the advantages respect to a batch reactor neither in the text nor in SI. They didn't report comparisons in batch conditions, please insert.**

Reply: The description of the continuous flow reactor system has been supplied in the Methods. As shown in Scheme S1, the continuous flow reactor system is composed of a stainless steel pipe with 750 mm length and 8 mm internal diameter. In addition, we compare the catalytic performance of the selective hydrogenation of BYD over as-prepared Pd/ZnO-400 catalyst in the continuous flow reactor and the batch reactor. As shown in Figure S9, the as-prepared Pd/ZnO-400 catalyst also presented highly efficient in the selectivity hydrogenation of BYD to *cis*-BED in a batch reactor. High selectivity to *cis*-BED (97.3%) at the conversion of 99.5% can be achieved at 2.0 MPa H₂ and 80 °C for 2.0 h. However, with extending the reaction time to 5.0 h, the over-hydrogenation of *cis*-BED was carried out. The selectivity of *cis*-BED slightly

decreased to 94.1% while the selectivity to BDO reached to 4.8%. Compared with the selectivity hydrogenation of BYD in a batch reactor, the products obtained from the reaction in the continuous flow reactor was high stability at the fixed reaction conditions. In addition, it was adopted for industrial scale-up through the selectivity hydrogenation of BYD in the continuous flow reactor.

Modification:

➤ In the revised manuscript

As a comparison, the hydrogenation of BYD over Pd/ZnO-400 catalyst was also evaluated in a batch reactor. As shown in Figure S9, the Pd/ZnO-400 catalyst also presented high efficiency in the selectivity hydrogenation of BYD to *cis*-BED. However, the products obtained from the reaction in the continuous flow reactor had a higher stability at the fixed reaction conditions than in the batch reactor. In addition, it is usually adopted for industrial scale-up through the selectivity hydrogenation of BYD in the continuous flow reactor.

The liquid-phase selective hydrogenation of alkynols over Pd/ZnO-*T* samples, Pd/Al₂O₃, and the commercial Pd-based catalysts (Pd/C, Pd/TiO₂, Pd/SiO₂, Pd/CaCO₃, PdAg/Al₂O₃, Lindlar) were performed in a continuous flow reactor system consists of a stainless steel pipe (length, 750 mm; internal diameter, 8 mm) (as shown in Scheme S1). As comparison, the BYD hydrogenation performance over Pd/ZnO-400 catalyst in the batch reactor was also carried out (The details are described in SI).

➤ In the supporting information

4. Hydrogenation of BYD in the batch reactor.

Liquid-phase hydrogenation of BYD was carried out in a 50 mL stainless autoclave containing 0.1 g as-prepared Pd/ZnO catalyst, 25 mL of feedstock with 5 wt.% BYD, 1 wt.% 1,2-propylene glycol (as internal standard) and 96% water (as solvent) and 2.0 MPa H₂ at 80 °C. The reaction system was stirred vigorously (700 rpm) to eliminate the diffusion effect. Liquid aliquots were taken periodically using a dip tube. The products were analyzed by gas chromatography (Agilent 7890B-GC).

Scheme S1 The unit of high pressure fixed bed continuous flow stainless steel catalytic reactor.

Figure S9 (a) The variation of the relative concentration and (b) the products

selectivities with reaction time for the hydrogenation of BYD over Pd/ZnO-400 catalyst in a batch reactor versus time (Reaction conditions: 0.1 g catalyst, reaction temperature 80 °C, 2 MPa H₂, and 25 g 5 wt.% BYD aqueous solution).

• Tables with complete data of conversions, selectivity, STY, TOF, TON, productivity, should be added for hydrogenation of 2-butyn-1,4-diol but also for 2-methyl-3-butyn-2-ol, 3-hexynol, 3-phenyl-2-propyn-1-ol, described at page 18 only with TOFs.

Reply: The Table with complete data about the results of the semi-hydrogenation of industrially relevant alkynols over Pd/ZnO-400 catalyst has been added in the revised manuscript.

Modification:

As presented in Table 1, the TOFs for the hydrogenation of these alkynols over Pd/ZnO-400 catalyst was follows: 2-methyl-3-butyn-2-ol (397 min⁻¹) > 3-hexyn-1-ol (157 min⁻¹) > 3-phenyl-2-propyn-1-ol (110 min⁻¹) > BYD (71 min⁻¹).

Table 1 The semi-hydrogenation of industrially relevant alkynols over Pd/ZnO-400 catalyst

Entry	Substrate	Product	Solvent	T (°C)	P (MPa)	τ^a (g _{cat} ·h·mol ⁻¹)	Conv. (%)	Sel. (%)	TOFs ^b (min ⁻¹)
1			water	80	2	29	95.8	92.6	71
2			methanol	100	3	67	88.5	90.0	110
3			methanol	130	3	41	97.7	97.2	157
4			water	50	1	29	98.5	95.8	397

^a τ meant the contact time.

^b The TOFs was calculated with the initial conversion of substrate <30%.

• The authors report only the contact time but it's not clear if they change only the flow of substrate, maintaining the content of catalyst (0.100 g ???) or both. In a flow process It's very important to show how productivity changes on the basis of parameters such as Hydrogen-flow, substrate-flow, time-contact, etc. Please specify.

Reply: The contact time is defined as the ratio of catalyst mass to molar flow rate of reaction substrate. In our experiment, the contact time has been changed by tuning the flow of substrate with maintaining the content of catalyst (0.1 g). For understanding the variation of productivity on the basis of parameters (reaction temperature, H₂ pressure, and the ratio of H₂/substrate), the related experimental data has been added.

Modification: The concept of contact time has been added in the experimental section of revised manuscript.

$$\text{Contact time (g}_{\text{cat.}} \text{ h mol}^{-1}) = m_{\text{cat.}} / (F * 60) \quad (1)$$

where $m_{\text{cat.}}$ (g) was the mass of catalyst (0.1 g), F (mol min⁻¹) was the molar flow rate of reactant.

• TOFs values obtained are good an in line with literature data but using hard reaction conditions in term of Pressure (2MPa), temperature (80°C), H₂/substrate ratio=600.

Reply: We investigate the selective hydrogenation of BYD over Pd/ZnO catalyst at mild conditions. As shown in Table R1, the TOF value indeed increases with increasing the reaction temperature and H₂ pressure. The hard reaction conditions in

term of pressure and temperature promote the BYD adsorption/activation on the surface of Pd/ZnO-400 catalyst, improving the TOF value to some extent. In our future research, the high TOFs for the hydrogenation of alkynols at mild conditions will be pursued.

Table R1 The effect of reaction conditions on the TOF values.

sample	T (°C)	P (MPa)	H ₂ /substrate ratio	TOF ^a (min ⁻¹)
Pd/ZnO-400	80	2	600	71
Pd/ZnO-400	80	0.1	600	56
Pd/ZnO-400	RT	2	600	37

^a Apparent TOF was calculated as moles of converted BYD per mole of Pd per minute with the relative low conversion.

• **I suggest you to add experiments at different temperatures (e.g. room temperature), low pressures (e.g. 1-2 bar) and low H₂/Substrate ratio, changing the hydrogen flow. This type of reactions doesn't need hard conditions. Mild conditions favour selectivity.**

Reply: According your suggestion, we added the experiments to explore the effects of reaction temperatures, H₂ pressures, and H₂/substrate ratios on the catalytic performances in the selectivity hydrogenation of BYD over Pd/ZnO-400 catalyst. Indeed, the selective hydrogenation of BYD to *cis*-BED is suitable under mild conditions. High selectivity to *cis*-BED (>97%) at almost complete conversion of BYD can be achieved at 80 °C, 0.5 MPa, and 100 H₂/substrate ratio.

Modification:

➤ In the revised manuscript

The effects of reaction temperatures, H₂ pressures, and the H₂/substrate volume ratios on the catalytic performances in the selectivity hydrogenation of BYD over Pd/ZnO-400 catalyst in the continuous flow reactor were also explored, and the results were shown in Figure S10-S12. The selective hydrogenation of BYD to *cis*-BED was suitable under mild conditions. High selectivity to *cis*-BED (>97%) at almost complete conversion of BYD can be achieved at 80 °C, 0.5 MPa, and 100 H₂/substrate ratio. Although the conversion of BYD was boosted under the hard conditions, it easily led to the formation of by-products and lowering the selectivity to intermediate *cis*-BED. Figure S10 showed that *cis*-BED was prone to isomerization and over-hydrogenation at high temperature, while the over-hydrogenation of C=C bond to C-C bond and the cleavage of C-OH bond were enhanced under high H₂ pressure (as shown in Figure S11). Additionally, the selective hydrogenation of BYD to *cis*-BED in the continuous flow reactor can be operated at the relative low H₂/substrate volume ratio with 100 from the perspective of hydrogen economics (Figure S12) and it may be necessary to recycle hydrogen in the industrial scale-up.

The effects of reaction temperatures (RT-100 °C), H₂ pressures (0.1-2.0 MPa), the volume ratio of H₂-to-liquid feed (50-600), and the contact time (0-87 g_{cat.}·h·mol⁻¹) on the catalytic performances in the selectivity hydrogenation of BYD over Pd/ZnO-400 catalyst were explored.

➤ In the supporting information

Figure S10 The effect of reaction temperature on the selective hydrogenation of BYD over Pd/ZnO-400 catalyst (Reaction conditions: 0.1 g catalyst, 2 MPa H₂, contact time 29 g_{cat.}·h·mol⁻¹, and the volume ratio of H₂ to liquid feed 600).

Figure S11 The effect of H₂ pressure on the selective hydrogenation of BYD over Pd/ZnO-400 catalyst (Reaction conditions: 0.1 g catalyst, reaction temperature 80 °C, contact time 29 g_{cat.}·h·mol⁻¹, and the volume ratio of H₂ to liquid feed 600).

Figure S12 The effect of the volume ratio of H₂ to liquid feed on the selective hydrogenation of BYD over Pd/ZnO-400 catalyst (Reaction conditions: 0.1 g catalyst, reaction temperature 80 °C, 2 MPa H₂, and contact time 29 g_{cat}·h·mol⁻¹).

• **Authors compare their catalyst only with Lindlar catalyst and Pd/Al₂O₃. Please add comparison with other commercial catalysts (e.g. Pd/C; Pd/TiO₂; Pd/SiO₂; Pd/CaCO₃) in the same reaction conditions.**

Reply: Thank you very much for your suggestion. Some other commercial Pd-based catalysts including Pd/C, Pd/TiO₂, Pd/SiO₂, and Pd/CaCO₃ have been evaluated in the selective hydrogenation of BYD. The conversion of BYD and the products distribution are displayed in Figure 3(b). Under the same reaction conditions, the commercial monometallic Pd-based catalysts present high activity and achieve 100% conversion of BYD while the product distributions are relative complex with relative low selectivity to *cis*-BED. The over-hydrogenation and *cis-trans* BED isomerization is prone to carry out. It further confirms that the Pd/ZnO-400 catalyst contained PdZn as active sites presents a promising application prospects in the selectivity hydrogenation

of BYD to *cis*-BED.

Modification: Figure 3(b) has been changed and the related discussion has been added in the revised manuscript.

Under the same reaction conditions, 100% conversion of BYD can be achieved over monometallic Pd-based catalysts, but the product distributions are relatively complicated. The selectivity to *cis*-BED was only 7.5% for Pd/Al₂O₃, 1.5% for Pd/SiO₂, 0.4% for Pd/C, 19.3% for Pd/TiO₂, and 94.3% for Pd/CaCO₃, respectively. The over-hydrogenation of *cis*-BED was prone to carry out over monometallic Pd-based catalysts, especially for the Pd/SiO₂ and Pd/C catalysts with weak metal-support interaction. In addition, the hydrogen-mediated *cis-trans* BED isomerization was observed on Pd/TiO₂ catalyst and the selectivity to *trans*-BED reaches 28.9%. This phenomenon was similar with the selective stereochemical catalysis over RhSb/SiO₂ [41].

The liquid-phase selective hydrogenation of alkynols over Pd/ZnO-*T* samples, Pd/Al₂O₃, and the commercial Pd-based catalysts (Pd/C, Pd/TiO₂, Pd/SiO₂, Pd/CaCO₃, PdAg/Al₂O₃, Lindlar) were performed in a continuous flow reactor system consists of a stainless steel pipe (length, 750 mm; internal diameter, 8 mm) (as shown in Scheme S1).

Figure 3. Selective hydrogenation properties of catalysts. (a) Reaction pathway for the hydrogenation of BYD. (b) Conversion of BYD and product selectivity over Pd/ZnO-*T* catalysts and the commercial Pd-based catalysts (Reaction conditions: 2 MPa H₂, 80 °C, and contact time 29 g_{cat.} h mol⁻¹). (c-d) Product distribution and selectivity of the BYD hydrogenation versus the contact time over Pd/ZnO-400 catalyst. (e) The stability testing of Pd/ZnO-400 catalyst in the BYD hydrogenation, in which the catalyst is regenerated at 110 h. (f) XPS spectra of Pd 3d region of the fresh and spent Pd/ZnO-400 catalyst.

[41]Furukawa, S., Ochi, K., Luo, H., Miyazaki, M. & Komatsu, T. Selective stereochemical catalysis controlled by specific atomic arrangement of ordered alloys. *ChemCatChem* **7**, 3472–3479, (2015).

• **Please describe better in situ regeneration by H₂, reaction conditions and characterization of catalyst fresh, used and regenerated.**

Reply: In the stability testing, the Pd/ZnO-400 catalyst was *in situ* regeneration by reducing it under 48 sccm hydrogen flow at 400 °C for 2 h before terminating the liquid feedstock. After in situ regeneration, the second stage stability testing was carried out at the same reaction conditions (2 MPa H₂, 80 °C, and contact time 29 g_{cat}. h mol⁻¹). The spent Pd/ZnO-400 catalyst was characterized by TG and XPS for understanding its structure.

Modification:

After terminating the liquid feedstock and *in situ* reducing the catalyst under 48 sccm hydrogen flow at 400 °C for 2 h, the second stage stability test was carried out at the same reaction conditions.

TG/DTG analysis of the PdO/ZnO sample and the spent Pd/ZnO-400 catalyst was performed on a Mettler-Toledo TGA/SDTA3+ jointing with Pfeiffer OmniStar mass spectrometer.

The XPS measurements were performed on Thermo VG Scientific ESCALAB250 electron spectrometers using a monochromatic Al K α X-ray source. All Pd/ZnO-T samples were pre-reduced at different temperatures before test and then transferred

into the chamber by the Schlenk technology while the spent Pd/ZnO-400 catalyst was directly moved into the chamber for the characterization.

• **No information about leaching in solution was considered. Please measure metal leaching in solution by ICP.**

Reply: According to your suggestion, the content of metals in the reaction solution has been detected by ICP-AES. There is no leaching of Pd and Zn in solution, which may be below the detection limit. The related description has been added in the revised manuscript.

Modification:

Through analyzing the content of metals in the reaction solution by ICP-AES, no leaching of Pd and Zn was observed (below the detection limit).

The practical loading of Pd in the samples and the content of Pd and Zn in the reaction solution were analyzed by ICP-AES.

5) the article doesn't offer strong element of novelty both in term of materials and catalysis, already described in literature and not sufficiently cited (e.g. among them: for materials: ACS catal 2017,7,2, 1491; Catal. Today 2018, 330, 2; J. Phys. Chem C 2011,115, 8457; ACS Appl. Nanomaterials 2019, 2,5,3307; for Hydrogenation of alkynes: Chem Eng Proc. Proc Intensification 2020, 154, 108018; ACS Sust Chem Eng 2019, 7, 13, 11050; J. Catal, 2014, 311, 212; Beilstein. J. Org Chem 2017, 13, 734; ChemCatChem 2017, 9, 3245). Please add some references for both arguments.

Reply: Thank you very much for your comments. Designing highly active and stable

Pb-free Pd-based catalysts without introducing of surfactants and stabilizers is vital for large-scale and efficient production of *cis*-enols through continuous-flow hydrogenation of alkynols, which is a critical process in the fine and intermediate chemical industry. In the past decade, some related studies on the continuous-flow processes for the catalytic partial hydrogenation of alkynols over Pd-based catalysts have emerged. The activity of the related catalysts as well as their stability and selectivity for target products were shown to be determined by the intrinsic metal active sites, promoters (second metals, organic ligands, and stabilizers), and the nature of supports (pore structure, acid- base properties, and SMSI). Compared with traditional Lindlar- type catalysts, such Pd- based intermetallics can be extensively used for the selective hydrogenation of alkynols, as such doping enhances selectivity for enols while avoiding the hazards associated with toxic additives. However, the related research is lack, although the Pd-based intermetallics catalysts for the semi-hydrogenation of acetylene are widely focused on and researched. In this work, the finely surface engineering technology in oxide-supported metal (particles) has been applied to controllable tune the geometric and electronic structure of Pd active sites for achieving the directional hydrogenation of alkynols. Though coupling strategy of strong electrostatic adsorption and reactive metal-support interaction, intermetallic PdZn/ZnO catalyst is designed. Driven by hydrogen at high temperature, the formation of intermetallic PdZn nanoparticles with electron-poor ($\text{Pd}^{\delta+}$) active sites on a ZnO supported Pd catalyst significantly boosts the thermodynamic selectivity with respect to the mechanistic selectivity and therefore enhances the

selectivity to *cis*-enols. The optimized Pd/ZnO-400 catalyst exhibits superior activity, unique selectivity, and long term stability in the continuous-flow semi-hydrogenation of alkynols. Based on the *in situ* DRIFTS and simulation calculation, the semi-hydrogenation of alkynols over intermetallic PdZn/ZnO has been proposed. Overall, on the basis of previous studies, this work is innovative in the design of intermetallic compound catalyst, the optimization of reaction process, and the understanding of reaction mechanism.

Modification: The related references have been added in the revised manuscript.

Owing to the large market demand for enols, continuous-flow semi-hydrogenation of alkynols over heterogeneous catalysts will be an inevitable trend to obtain high-quality products under large-scale applications in virtue of their process safety, environmental friendliness, and economic benefits [5-10].

Intermetallic compounds (IMCs) with ordered crystal structures and special electronic structure play a key role considering the practical manufacturing of heterogeneous catalysts [14-17].

The introduction of a second metal tunes the ensemble and ligand effects of metallic Pd significantly, enhancing the selective hydrogenation of alkynes [24-26].

[5] Liguori, F. & Barbaro, P. Green semi-hydrogenation of alkynes by Pd@borate monolith catalysts under continuous flow. *J. Catal.* **311**, 212–220 (2014).

[6] Tanielyan, S. K., More, S. R., Augustine, R. L. & Schmidt, S. R. Continuous liquid-phase hydrogenation of 1,4-butyne-1,3-diol to high-purity 1,4-butanediol over particulate Raney nickel catalyst in a fixed bed reactor. *Org. Process Res. Dev.* **21**,

327–335 (2017).

[7] Moreno-Marrodan, C., Liguori, F. & Barbaro, P. Continuous-flow processes for the catalytic partial hydrogenation reaction of alkynes. *Beilstein J. Org. Chem.* **13**, 734–754 (2017).

[8] Liguori, F. et al. Unconventional Pd@sulfonated silica monoliths catalysts for selective partial hydrogenation reactions under continuous flow. *ChemCatChem* **9**, 3245–3258 (2017).

[9] Fukazawa, A. et al. A new approach to stereoselective electrocatalytic semihydrogenation of alkynes to Z- alkenes using a proton-exchange membrane reactor. *ACS Sustainable Chem. Eng.* **7**, 11050–11055 (2019).

[10]Kundra, M. et al. Continuous flow semi-hydrogenation of alkynes using 3D printed catalytic static mixers. *Chem. Eng. Process.* **154**, 108018 (2020).

[16]Dasgupta, A. & Rioux, R. M. Intermetallics in catalysis: An exciting subset of multimetallic catalysts. *Catal. Today* **330**, 2–15 (2019).

[24]Tew, M. W., Emerich, H. & van Bokhoven, J. A. Formation and characterization of PdZn Alloy: A very selective catalyst for alkyne semihydrogenation. *J. Phys. Chem. C* **115**, 8457–8465 (2011).

[25]Pei, G. X., et al. Performance of Cu-Alloyed Pd single-atom catalyst for semihydrogenation of acetylene under simulated front-end conditions. *ACS Catal.* **7**, 1491–1500 (2017).

[26]Miyazaki, M., Furukawa, S., Takayama, T., Yamazoe, S. & Komatsu, T. Surface modification of PdZn nanoparticles via galvanic replacement for the selective

hydrogenation of terminal alkynes. *ACS Appl. Nano Mater.* **2**, 3307–3314 (2019).

6) The suppression of overhydrogenation of BED to BDO, and the selectivity to cis-BED, cited at page 16, can be explained by DFT analysis at page 22 ?? Explain better. It's not clear. In case, please add in the text, at page 16, some reference to this part .

Reply: Indeed, the subsequent DFT analysis can explain the suppression the over-hydrogenation of BED to product of BDO over intermetallic PdZn particles. For better understanding this phenomenon, the related research work on the semi-hydrogenation of acetylene over ultra-small intermetallic PdZn particles has been cited in the revised manuscript.

Modification:

The SMSI effect with respect to Pd-Zn covalent interaction partially blocked the non-selective active sites of Pd NPs enhances the adsorption of the BYD with respect to that of the *cis*-BED, thus suppressing the over-hydrogenation of BED to product of BDO. This was similar with that the energetically favorable path for acetylene hydrogenation and ethylene desorption over the ultra-small intermetallic PdZn particles [43].

[43]Hu, M. et al. MOF-confined sub-2 nm atomically ordered intermetallic PdZn nanoparticles as high-performance catalysts for selective hydrogenation of acetylene. *Adv. Mater.* **30**, 1801878 (2018).

7) Why do you make DRIFT spectroscopy with propargyl alcohol ? Why do you use such a danger /toxic compound, never used in previous cited catalytic

experiments? Why don't you use 2-butyne-1,4-diol or others?

Reply: That is a good question. The previous researches on the hydrogenation of acrolein and α - amino ester probed by FTIR spectroscopy provide a great guide on the explore the hydrogenation of alkynol by applying in situ DRIFT spectroscopy [1-3]. In our research, the *in situ* DRIFT spectroscopy with propargyl alcohol has been utilized for identifying the reactive surface intermediates in the selective hydrogenation of alkynol over PdZn/ZnO catalyst. Generally, both alkyne and C-OH group are presented in propargyl alcohol, which has a similar structure with 2-butyne-1,4-diol. It can be as a candidate to explore the mechanism of selective hydrogenation of C \equiv C bonds to C=C bonds and the possible hydrogenolysis or hydrodeoxygenation of C-OH bonds through DRIFT spectroscopy. In addition, the boiling point of propargyl alcohol (114.5 °C) is relative lower than that of 2-butyne-1,4-diol (228 °C). It is very convenient to feed into the Praying Manitis DRIFTS reactor under the action of Ar as carrier gas to detect the adsorption or hydrogenation over Pd/ZnO sample.

References:

1. Dostert, K.-H., O'Brien, C. P., Ivars-Barceló, F., Schauer mann, S. & Freund, H.-J. Spectators control selectivity in surface chemistry: Acrolein partial hydrogenation over Pd. *J. Am. Chem. Soc.* **137**, 13496–13502 (2017).
2. Dostert, K.-H. et al. Selective partial hydrogenation of acrolein on Pd: A mechanistic study. *ACS Catal.* **7**, 5523–5533 (2017).
3. Zhang, L., Lohrasbi, M., Tumuluri, U. & Chuang, S. S. C. Asymmetric hydrogenation of α - amino ester probed by FTIR spectroscopy. *Org. Process Res.*

Dev. **20**, 1668–1676 (2016).

8) DFT calculations are reported at the beginning (page 5-7) and then at the end of the paper (page 22-23) with reference to figures and schemes at the beginning and to SI. It makes difficult to read and to understand the reasoning. I suggest to write only a chapter about DFT calculation at the end of the paper to explain all details for thermodynamics and mechanistic selectivity. In general, for a better understanding the text, I suggest to follow a scheme: synthesis of catalyst, complete characterization of catalyst, catalytic experiments in flow and batch, DFT calculations and DRIFT spectroscopy to explain the selectivity and mechanism, conclusion, methods. Now everything it's mixed and confusing.

Reply: Thank you very much for your suggestion. For better understanding the text, the structure of the article has been re-adjusted. Now, we wrote the Results in the following order: PdZn/ZnO catalyst prepared by SEA and following reduction, Selective hydrogenation of alkynols, *In situ* DRIFT spectroscopy of the hydrogenation of propargyl alcohol, DFT analysis of reaction mechanism. In addition, the descriptions and the Figures has been reconstructed. Some Figures have been transferred in the supporting information.

Modification:

➤ In the revised manuscript

PdZn/ZnO catalyst prepared by SEA and following reduction. Site modification and isolation was known to be a powerful strategy of enhancing the selectivity of Pd-based catalysts for the semi-hydrogenation of alkynols to enols [32].

Here, a PdZn/ZnO catalyst have been designed by the relative coupling strategy of SEA and following high temperature reduction, in which Pd active sties were modified by Zn *via* formation of intermetallic PdZn, as shown in Figure 1(a) [33]. Due to the strong covalent Pd–Zn interaction, the pronounced charge correlation regions around the Pd atom owing to electron accumulation can be observed in Figure 1(b). The calculated results showed that the Mulliken atomic charges are ca. -0.17 and 0.18 e for the superficial Pd and Zn atom, respectively. Based on the fact that the *d*-band width of metal directly effected its capacity to adsorb unsaturated chemical bonds by the semi-empirical method [34], the intermetallic PdZn/ZnO catalyst may present high potential in the selective hydrogenation of alkynols.

DFT analysis of reaction mechanism. To better understand the underlying mechanism, the calculated total electronic density of states of BYD, BED, Pd, and PdZn and the adsorption configurations and energies of reaction species over Pd (111) and PdZn (111) were calculated. As shown in Figure S16, the *d* bands center of intermetallic PdZn presented a negative deviation and the bandwidth was reduced significant in comparison with metallic Pd, which may be ascribed to the missing *d-d* interactions and lattice strain, leading to the dramatic change in the catalytic properties [50]. Interestingly, the *d* bands of PdZn overlapped with the *p* bands of BYD slightly but there was almost no overlay with that of BED. However, they overlapped and interweaved among the *d* bands of metallic Pd with the *P* bands of BYD and BED. From the point of electronic effect, it can deduce that intermetallic PdZn boosts the selectivity to BED and suppresses its over-hydrogenation comparing

the metallic Pd due to the weaker interaction. Based on the thermodynamically stable and mostly exposed surfaces of PdZn (111) and Pd (111), the configurations and energies of the most stable BYD, *cis*-BED, and *trans*-BED adsorption on these surfaces were calculated, as shown in Figure S17. The adsorption energies of BYD on the PdZn (111) and Pd (111) were -0.97 eV and -1.30 eV, respectively, emerging highly efficient reactivity of Pd sites of the PdZn (111) and Pd (111). Moreover, the adsorption energies of *cis*-BED on the PdZn (111) and Pd (111) were -0.27 eV and -0.55 eV, respectively, which represented that the desorption of *cis*-BED on the surface of PdZn had priority over that on Pd. The same results can also be found for the *trans*-BED while the *cis-trans* BED isomerization should be further explored. In view of the above-mentioned facts, it can deduce that the thermodynamic selectivity has been enhanced and the over-hydrogenation or hydrogenolysis of BED over intermetallic PdZn may be inhibited significantly. Regulating the Pd active sites to the completely isolated by Zn given rise to preferential π - adsorption of C \equiv C bonds of BYD, and decreased the adsorption energies significantly, indicating the intermetallic PdZn is a promising catalyst for the semi-hydrogenation of BYD to BED.

Figure 1. Structure characterization of intermetallic PdZn. (a) Schematic illustrating PdZn/ZnO catalyst. (b) Deformation density of intermetallic PdZn. (c) TG/DTG profiles for the Pd/ZnO sample under 5 vol.% H₂/Ar from RT to 520 °C with 5 °C min⁻¹. (d) H₂-TPR profiles of Pd/ZnO and ZnO samples in 10 vol.% H₂/Ar up to 500 °C with 5 °C min⁻¹. (e) XPS spectra of Pd 3d region of unreduced PdO/ZnO and Pd/ZnO-*T* samples.

➤ In the revised manuscript

Figure S6. The molar ratio of PdZn/Pd versus the reduction temperature (PdO: navy blue, Pd: wine red, PdZn: yellow).

Figure S16 The calculated total electronic density of states of BYD, BED, Pd, and PdZn.

Figure S17 Adsorption configurations and energies of BYD, *cis*-BED, and *trans*-BED on PdZn (111) and Pd (111), respectively.

Reviewer #3:

The manuscript is structured logically and written in most parts in clear English, however in certain sections it needs to be rewritten substantially to meet the standard of Commschem. In these sections, the language, expressions or grammar used are inadequate and negatively impact comprehensibility, up to the point where certain statements are rendered unintelligible; especially the section named ‘Selective hydrogenation of alkynols’, p16ff, requires significantly more work before it can be accepted.

Reply: Thank you very much for your comments. According to your suggestion, the expression in English has been improved with the assistance of Dr. Chi-Wing Tsang, Technological and Higher Education Institute of Hong Kong.

Modification:

The language, expressions and grammar used in the manuscript have been modified, such as,

However, at high reduction temperatures, the reconstruction of catalyst as well as the formation of intermetallic PdZn **continued** to occupy center stage, which **enhanced** the ensemble and ligand effects on the Pd active sites, boosting the semi-hydrogenation of BYD to *cis*-BED.

It clearly showed that the selectivity to *cis*-BED can reach **to** ca. 92% with the BYD conversion of 96% over the as-prepared Pd/ZnO-400 catalyst.

The TOF value of Pd/ZnO-400 catalyst with the PdZn/Pd molar ratio 1:1 (71 min^{-1}) was slightly lower than that of Pd/ZnO-150 catalyst with pure Pd NPs supported on ZnO (90

min⁻¹), which further **indicated** that partial active sites **were** converged. Under the same reaction conditions, 100% conversion of BYD **can be achieved** over **monometallic Pd-based catalysts**, but the product distributions are **relatively complicated**.

Significantly, the selectivity to *cis*-BED over Pd/ZnO-400 catalysts maintained a relatively **stable production** with higher than 90.0% as contact time increased (Figure 3(d)). The SMSI effect with respect to Pd-Zn covalent interaction partially blocked the non-selective active sites of Pd NPs **preferred the thermodynamic selectivity comparing to the mechanistic selectivity**, thus suppressing the over-hydrogenation of *cis*-BED to product of BDO. **This was similar with the observation on the energetically favorable path for acetylene hydrogenation and ethylene desorption over the ultra-small intermetallic PdZn particles [43].**

It can be **deduced** that the non-selective PdH_x active sites with high activity in the surface of Pd/ZnO-400 catalyst **were** buried in oblivion as the extension of the reaction time.

In situ regeneration by H₂ reduction treatment is a **powerful** strategy to eliminate the strongly adsorbed species on the active sites and **strengthen** the interaction between inter(metallic) active center with oxides support [44].

1) Page 6, line 19: “Moreover, the adsorption energies of *cis*-BED on the PdZn (111) and Pd (111) are -0.27 eV and -0.55 eV, respectively...” – what about the adsorption energies of *trans*-BED? In the experimental hydrogenation section the authors have found that the alkene predominantly produced is *cis* not *trans*,

which agrees with most literature. Nevertheless, information on the adsorption energies of trans-BED might be useful here to complete the picture.

Reply: Thank you very much for your suggestion. The adsorption energies of *trans*-BED on the PdZn (111) and Pd (111), respectively, have been calculated. As shown in Figure S18, the adsorption energies of *trans*-BED on the PdZn (111) and Pd (111) are -0.27 eV and -0.61 eV, respectively, which are almost similar with the adsorption energies of *cis*-BED on the PdZn (111) and Pd (111). Just considering the adsorption energy, it is hardly to clarify that the selective hydrogenation of BYD predominantly produced is *cis*-BED not *trans*-BED. However, the energy distribution diagrams of the hydrogenation pathways of C≡C bonds in BYD (Figure 5) present that the hydrogenation barriers of *trans*-TS_{2Pd} is close to that of *cis*-TS_{2Pd}, indicating that *cis/trans*-isomerism of BED can be obtained over metallic Pd catalyst. While the value of *cis*-TS_{2PdZn} is much lower than that of *trans*-TS_{2PdZn}, which indicates that there is extremely unlikely to carry out the *cis/trans*-isomerism of BED over intermetallic PdZn.

Modification:

➤ In the revised manuscript

Based on the thermodynamically stable and mostly exposed surfaces of PdZn (111) and Pd (111), the configurations and energies of the most stable BYD, *cis*-BED, and *trans*-BED adsorption on these surfaces were calculated, as shown in Figure S17. The adsorption energies of BYD on the PdZn (111) and Pd (111) were -0.97 eV and -1.30

eV, respectively, emerging highly efficient reactivity of Pd sites of the PdZn (111) and Pd (111). Moreover, the adsorption energies of *cis*-BED on the PdZn (111) and Pd (111) were -0.27 eV and -0.55 eV, respectively, which represented that the desorption of *cis*-BED on the surface of PdZn had priority over that on Pd. The same results can also be found for the *trans*-BED while the *cis-trans* BED isomerization should be further explored. In view of the above-mentioned facts, it can deduce that the thermodynamic selectivity has been enhanced and the over-hydrogenation or hydrogenolysis of BED over intermetallic PdZn may be inhibited significantly.

➤ In the Supporting Information

The adsorption configurations and energies of BYD, *cis*-BED, and *trans*-BED on PdZn (111) and Pd (111), respectively, have been shown in Supporting information (Figure S17).

Figure S17 Adsorption configurations and energies of BYD, *cis*-BED, and *trans*-BED on PdZn (111) and Pd (111), respectively.

2) Page 8, line 11: The term ZnO-*T* should be introduced here. This is done later on Page 26:"The obtained catalysts were assigned as Pd/ZnO-*T*, where *T* referred to the reduction temperatures. " - this should be done here.

Reply: Thank you very much for pointing this mistake. The physical meaning of *T* should be assigned at first mention. In the revised manuscript, we have introduced it at page 7 and removed it in section of Methods.

Modification:

As shown in Figure S2, XRD patterns of ZnO and Pd/ZnO-*T* samples (*T* referred to the reduction temperatures) only presented the typical diffraction peaks of ZnO, indicating that the well-dispersed Pd NPs on the ZnO support are too small to be detected.

3) Page 9, figure 2: The diagram 2d is not clear. How do the differently coloured images of the catalyst correlate to the graph? The link needs to be explained.

Reply: As show in Figure 2d (displayed in Figure S6 after revised), the molar ratio of PdZn/Pd calculated based on the XPS peak area (Figure 1e) has been related to the reduction temperature. In order to understand the configuration of the Pd/ZnO-*T* catalyst, the scheme of catalysts has been described. The PdO, Pd, and PdZn nanoparticles have been shown in navy blue, wine red, and yellow images, respectively.

Modification: In the supporting information, the mean of colored images (as shown in Figure S6) has been added.

Figure S6. The molar ratio of PdZn/Pd versus the reduction temperature (PdO: navy blue, Pd: wine red, PdZn: yellow).

4) Page 10, line 19: “Ultimately, in the reduction process, intermetallic PdZn NPs are formed through two-step processes including the reduction of PdO NPs as well as ZnO support partially and subsequently the diffusion of Zn atoms into Pd lattices”, What proof is there for migration of Zn into the Pd lattice?

Reply: From the TG/DTG-MS measurements of Pd/ZnO sample in H₂ atmosphere and H₂-TPR, we can only infer that the formation of PdZn intermetallic compound due to the SMSI effect. However, it provides the directly evidence to proof the formation of PdZn intermetallic compound for the Pd/ZnO-400 sample and the Zn in the Pd lattice by the XPS and HAADF-STEM measurements.

Modification:

Ultimately, in the reduction process, intermetallic PdZn NPs are formed through two-step processes including the reduction of PdO NPs as well as ZnO support

partially and subsequently the diffusion of Zn atoms into Pd lattices due to the SMSI effect, which will be confirmed by subsequent X-ray photoelectron spectroscopy (XPS) and high-angle annular dark-field scanning transmission electron microscopy (HAADF-STEM) measurements.

5) Page 12, figure 3: It is not clear from the marking on the TEM images how distance measurements were conducted. What's the double dashed line on a), b) and c)?

Reply: The HAADF-STEM images of Pd/ZnO-400 sample have been analyzed by GMS digital micrograph software. Through the tool of Profile, We can get a rectangular box perpendicular to lattice stripe, as shown in Figure R1(a). In the Profile window (as shown in Figure R1(b) and (c)), a dotted line box can be dragged out and the distance is displayed. Finally, the crystal plane index of sample can be obtained by comparing the crystal plane spacing of XRD standard spectrum. In our manuscript, the double dashed lines on Figure 2(b-c) have been used to label the lattice stripes.

Figure R1 The analysis of HRTEM image of Pd/ZnO-400 sample by GMS digital

micrograph software.

6) Page 13, line 20: “The catalytic performance of SEA-prepared Pd/ZnO-*T* catalysts in 5 wt.% BYD aqueous system was evaluated at 2 MPa H₂, 80 °C, and contact time 29 g_{cat.}·h·mol⁻¹.” How was the contact time derived?

Reply: The contact time is defined as the ratio of catalyst mass to molar flow rate of reaction substrate.

Modification: The concept of contact time has been added in the experimental section of revised manuscript.

$$\text{Contact time (g}_{\text{cat.}} \cdot \text{h mol}^{-1}) = m_{\text{cat.}} / (F \cdot 60) \quad (1)$$

where $m_{\text{cat.}}$ (g) was the mass of catalyst (0.1 g), F (mol min⁻¹) was the molar flow rate of reactant.

7) Page 15, figure 4: “(a) Conversion of BYD and product selectivity over Pd-based catalysts (Reaction conditions: 2 MPa H₂, 80 °C, and contact time 29 g_{cat.} h mol⁻¹).” – The bar graphs in figure 4a) show performance of catalysts prepared under different reduction temperatures; this is not made clear in the caption and also higher up in the text (end of page 13) and therefore the description should be altered to avoid confusion.

Reply: Both the as-prepared Pd/ZnO-*T* catalysts and the commercial Pd-based catalysts have been evaluated in the selective hydrogenation of BYD under 2 MPa H₂, 80 °C, and contact time 29 g_{cat.} h mol⁻¹. For avoiding confusion, the caption of Figure 3b has been changed into “Conversion of BYD and product selectivity over Pd/ZnO-*T*

catalysts and the commercial Pd-based catalysts (Reaction conditions: 2 MPa H₂, 80 °C, and contact time 29 g_{cat.} h mol⁻¹)”

Modification: The caption of Figure 3b has been modified in the revised manuscript.

Figure 3. Selective hydrogenation properties of catalysts. (a) Reaction pathway for the hydrogenation of BYD. (b) Conversion of BYD and product selectivity over Pd/ZnO-*T* catalysts and the commercial Pd-based catalysts (Reaction conditions: 2 MPa H₂, 80 °C, and contact time 29 g_{cat.} h mol⁻¹). (c-d) Product distribution and

selectivity of the BYD hydrogenation versus the contact time over Pd/ZnO-400 catalyst. (e) The stability testing of Pd/ZnO-400 catalyst in the BYD hydrogenation, in which the catalyst is regenerated at 110 h. (f) XPS spectra of Pd 3d region of the fresh and spent Pd/ZnO-400 catalyst.

8) Page 16ff: “Significantly, the selectivity to cis-BED over Pd/ZnO-400 catalysts maintains a relatively stabilization with higher than 90.0% with increasing contact time (Figure 4(c)). The SMSI effect with respect to Pd-Zn covalent interaction partially blocked the non-selective active sites of Pd NPs enhances the adsorption of the BYD with respect to that of the cis-BED, thus suppressing the over-hydrogenation of BED to product of BDO.”, this section and following sections need to be rewritten to make it clear what the authors would like to say.

Reply: To void the confusion, these sections have been rewritten. The Pd/ZnO-400 catalyst with intermetallic PdZn active sites presents high efficient and high selectivity to *cis*-BED in the hydrogenation of BYD, which may be attributed that the formation of Pd-Zn covalent interaction partially blocks the non-selective active sites of Pd NPs, preferring to the thermodynamic selectivity comparing to the mechanistic selectivity. Therefore, the over-hydrogenation of *cis*-BED to BDO has been inhibited. The same phenomenon was also found in the semi-hydrogenation of acetylene over the ultra-small intermetallic PdZn particles. The related reference has been added in the revised manuscript.

Modification:

The SMSI effect with respect to Pd-Zn covalent interaction partially blocked the

non-selective active sites of Pd NPs preferred the thermodynamic selectivity comparing to the mechanistic selectivity, thus suppressing the over-hydrogenation of *cis*-BED to product of BDO. This was similar with the observation on the energetically favorable path for acetylene hydrogenation and ethylene desorption over the ultra-small intermetallic PdZn particles [43].

[43]Hu, M. et al. MOF-confined sub-2 nm atomically ordered intermetallic PdZn nanoparticles as high-performance catalysts for selective hydrogenation of acetylene. *Adv. Mater.* **30**, 1801878 (2018).

9) Page 17, line 11: “This is the first report about the long-term stability of Pd-based catalyst in a continuous-flow semi-hydrogenation of BYD to *cis*-BED.” This is not the first continuous flow study of Pd-based catalysts for the semi-hydrogenation of BYD to *cis*-BED! Very substantial work has been carried out over the past years on the semi-hydrogenation of this particular molecule, 2-butyne-1,4-diol. The authors should reference recent work on this reaction, especially in industry relevant continuous flow hydrogenation settings, such as the work from Moreno-Marrodan et al (*Beilstein J Org Chem.* 2017; 13: 734–754, doi: 10.3762/bjoc.13.73), Tanielyan et al (*Org. Process Res. Dev.* 2017, 21, 3, 327–335, <https://doi.org/10.1021/acs.oprd.6b00375>) and Kundra et al (*Chem. Eng. Process.* 2020, 154, 108018, <https://doi.org/10.1016/j.cep.2020.108018>).

Reply: Thank you very much for your comments. The sentence “This is the first report about the long-term stability of Pd-based catalyst in a continuous-flow semi-hydrogenation of BYD to *cis*-BED.” has been removed. The recent works on the

selective hydrogenation of BYD in continuous flow reactor have been supplied in the revised manuscript.

Modification:

Owing to the large market demand for enols, continuous-flow semi-hydrogenation of alkynols over heterogeneous catalysts will be an inevitable trend to obtain high-quality products under large-scale applications in virtue of their process safety, environmental friendliness, and economic benefits [5-10].

[5] Liguori, F. & Barbaro, P. Green semi-hydrogenation of alkynes by Pd@borate monolith catalysts under continuous flow. *J. Catal.* **311**, 212–220 (2014).

[6] Tanielyan, S. K., More, S. R., Augustine, R. L. & Schmidt, S. R. Continuous liquid-phase hydrogenation of 1,4-butyne-1,3-diol to high-purity 1,4-butanediol over particulate Raney nickel catalyst in a fixed bed reactor. *Org. Process Res. Dev.* **21**, 327–335 (2017).

[7] Moreno-Marrodan, C., Liguori, F. & Barbaro, P. Continuous-flow processes for the catalytic partial hydrogenation reaction of alkynes. *Beilstein J. Org. Chem.* **13**, 734–754 (2017).

[8] Liguori, F. et al. Unconventional Pd@sulfonated silica monoliths catalysts for selective partial hydrogenation reactions under continuous flow. *ChemCatChem* **9**, 3245–3258 (2017).

[9] Fukazawa, A. et al. A new approach to stereoselective electrocatalytic semihydrogenation of alkynes to Z-alkenes using a proton-exchange membrane reactor. *ACS Sustainable Chem. Eng.* **7**, 11050–11055 (2019).

[10]Kundra, M. et al. Continuous flow semi-hydrogenation of alkynes using 3D printed catalytic static mixers. *Chem. Eng. Process.* **154**, 108018 (2020).

10) Page 17, line 22: “weightlessness peak” – What’s a weightlessness peak, can the authors explain?

Reply: The weightlessness peak means the mass loss peak. The spent Pd/ZnO-400 catalyst has been detected by TG-DTG curve. As shown in Figure S8, there is no obvious mass loss peak in the high temperature region.

Modification: “weightlessness peak” has been replaced by “obvious mass loss peak” in the revised manuscript.

Additionally, it was noteworthy that no obvious mass loss peak was observed in the high-temperature region for the TG/DTG curve of spent Pd/ZnO-400 catalyst (as shown in Figure S13).

11) Page 20, section named ”In situ DRIFT spectroscopy of the hydrogenation of propargyl alcohol” – this section is very lengthy and parts of it should be moved to the supporting information.

Reply: Thank you very much for your comments. For shortening the section “In situ DRIFT”, some sentences have been deleted.

Modification:

The sentences “since the adsorption characteristics of propargyl alcohol on the catalyst surface are considered to be an important factor in determining its selective hydrogenation.” , “Comparing the results from Figure 4(b) and Figure S16, this side

reaction prefers to carry out over Pd/ZnO-150 sample.” have been deleted in the revised manuscript.

12) Page 28, line 8: “...and H₂-to-liquid feed ratio of 600.” – Is this feed ratio based on volume/volume or mass/mass?

Reply: The feed ratio is based on volume/volume in the manuscript.

Modification:

The effects of reaction temperatures (RT-100 °C), H₂ pressures (0.1-2.0 MPa), the volume ratio of H₂-to-liquid feed (50-600), and the contact time (0-87 g_{cat.}·h·mol⁻¹) on the catalytic performances in the selectivity hydrogenation of BYD over Pd/ZnO-400 catalyst were explored.

REVIEWERS' COMMENTS:

Reviewer #1 (Remarks to the Author):

After the modifications, this manuscript might be accepted for the publication.

Reviewer #2 (Remarks to the Author):

In my opinion, a more rigorous and detailed scientific analysis was proposed. In its revised form the manuscript was acceptable. The authors answered exhaustively and in a satisfactory way all my questions, so the article can be considered suitable for the publication.

Reviewer #3 (Remarks to the Author):

The authors have adequately addressed all questions and points raised during the initial review. The changes made to the manuscript have improved clarity and added valuable information. I can recommend for the revised manuscript to be accepted for publication in CommsChem.

Reviewers' comments:

Reviewer #1(Remarks to the Author):

This manuscript describes the design and preparation of highly active and stable Pb-free Pd-based catalysts for continuous-flow semi-hydrogenation of alkynols. Catalytic tests show that the optimized PdZn/ZnO-400 catalyst exhibits superior activity, unique selectivity, as well as outstanding stability in the continuous-flow semi-hydrogenation of alkynols to cis-enols, which can be as a credible alternative catalyst to the commercial PdAg/Al₂O₃ and Lindlar catalyst. These results are interesting, therefore I recommend to publish this work after minor revisions in the following:

1. As shown in Figure 2, authors should show details for XPS measurements. Were the tests performed at in-situ conditions or not?
2. Authors should show more details for the reaction mechanism, and I suggest to draw a proposed scheme.
3. Authors should give the Pd particle size distribution. Are there any smaller Pd particle size?

Reviewer #2 (Remarks to the Author):

The authors describe the synthesis of intermetallic PdZn/ZnO catalysts for the semihydrogenation of alkynols in flow reactor. The article doesn't offer strong element of novelty both in term of materials and catalysis, already described in

literature and not sufficiently cited. Although the article offers some interesting perspectives in my opinion it can't be considered suitable for the publication for several reasons, cited below. The manuscript is unacceptable in its present form. The study seems promising and the authors should be encouraged to consider resubmission in the future, probably in other journal with several review.

1) The manuscript is difficult to read and confusing. The lack of schemes of reaction, tables o schemes in the text, an overuse of acronyms (BID, BED, SMSI, RMSI IMC, SEA, DOS etc), an excessive reference to schemes in SI, make it hard to read . I suggest you to modify these points and to improve the English.

2) At pag 5, the author introduces the catalyst PdZn/ZnO-400 but he doesn't explain the meaning of "400". Only after we can deduce it's the "reduction temperature".

3) The synthesis of such catalyst was explained in part at the beginning (pag 7-8, with reference to Figure S1 in SI, and then again at the end (pag 25) with other details. In my opinion It's not so clear. Please explain better all details for the synthesis at the beginning or at the end.

4) The article offer an accurate characterization of catalysts but it's extremely lacking in term of data of catalysis (e.g. experiments of catalysis, parameters to evaluate the productivity of catalysts, an accurate description of the flow reactor, comparison with commercial catalysts).

- The authors use a flow reactor but they doesn't describe the instrument (Volume, length, materials, connections; check controls, process schemes) and the advantages

respect to a batch reactor neither in the text nor in SI. They didn't report comparisons in batch conditions, please insert.

- Tables with complete data of conversions, selectivity, STY, TOF, TON, productivity, should be added for hydrogenation of 2-butyne-1,4-diol but also for 2-methyl-3-butyne-2-ol, 3-hexynol, 3-phenyl-2-propyne-1-ol, described at pag 18 only with TOFs.
- The authors report only the contact time but it's not clear if they change only the flow of substrate, maintaining the content of catalyst (0.100 g ???) or both. In a flow process It's very important to show how productivity changes on the basis of parameters such as Hydrogen-flow, substrate-flow, time-contact, etc. Please specify.
- TOFs values obtained are good and in line with literature data but using hard reaction conditions in term of Pressure (2MPa), temperature (80°C), H₂/substrate ratio=600.
- I suggest you to add experiments at different temperatures (e.g. room temperature), low pressures (e.g. 1-2 bar) and low H₂/Substrate ratio, changing the hydrogen flow. This type of reactions doesn't need hard conditions. Mild conditions favour selectivity.
- Authors compare their catalyst only with Lindlar catalyst and Pd/ Al₂O₃ . Please add comparison with other commercial catalysts (e.g. Pd/C; Pd/TiO₂; Pd/SiO₂; Pd/CaCO₃) in the same reaction conditions.
- Please describe better in situ regeneration by H₂, reaction conditions and characterization of catalyst fresh, used and regenerated.

- No information about leaching in solution was considered. Please measure metal leaching in solution by ICP.

5) the article doesn't offer strong element of novelty both in term of materials and catalysis, already described in literature and not sufficiently cited (e.g. among them: for materials: ACS catal 2017,7,2, 1491; Catal. Today 2018, 330, 2; J. Phys. Chem C 2011,115, 8457; ACS Appl. Nanomaterials 2019, 2,5,3307; for Hydrogenation of alkynes: Chem Eng Proc. Proc Intensification 2020, 154, 108018; ACS Sust Chem Eng 2019, 7, 13, 11050; J. Catal, 2014, 311, 212; Beilstein. J. Org Chem 2017, 13, 734; ChemCatChem 2017, 9, 3245). Please add some references for both arguments.

6) The suppression of overhydrogenation of BED to BOD, and the selectivity to cis-BED, cited at pag 16, can be explained by DFT analysis at pag 22 ?? Explain better. It's not clear. In case, please add in the text, at pag 16, some reference to this part.

7) Why do you make DRIFT spectroscopy with propargyl alcohol? Why do you use such a danger /toxic compound, never used in previous cited catalytic experiments? Why don't you use 2-butyne-1,4-diol or others?

8) DFT calculations are reported at the beginning (pagg 5-7) and then at the end of the paper (pagg 22-23) with reference to figures and schemes at the beginning and to SI. It make difficult to read and to understand the reasoning. I suggest to write only a chapter about DFT calculation at the end of the paper to explain all details for thermodynamics and mechanistic selectivity. In general, for a better understanding the

text, I suggest to follow a scheme: synthesis of catalyst, complete characterization of catalyst, catalytic experiments in flow and batch, DFT calculations and DRIFT spectroscopy to explain the selectivity and mechanism, conclusion, methods. Now everything it's mixed and confusing.

Reviewer #3 (Remarks to the Author):

The manuscript entitled 'Hydrogen induced intermetallic PdZn for invoking the continuous-flow hydrogenation of alkynols to cis-enols' (COMMSCHEM-21-0260-T) by Chen et al. presents results from an experimental study on the preparation, characterisation and evaluation of new intermetallic PdZn catalysts for use in the selective semi-hydrogenation of 2-butyne-1,4-diol. The authors describe a targeted study of this new catalyst for the use in a special semi-hydrogenation where high selectivity towards the partially reduced alkene intermediate is desired and has significant industrial relevance. The industry standard for this reaction are Pb-containing noble metal catalysts such as Lindlar type catalysts, which have issues associated with their Pb content. In comparison the Pb-free catalyst the authors present here, even show improved selectivity compared to Lindlar catalysts. The authors' characterisation of this new PdZn catalyst is highly detailed and includes TG, X-ray spectroscopy methods such as XPS and EDX, TEM, DRIFT spectroscopy, and a DFT analysis of the reaction mechanism.

The manuscript is structured logically and written in most parts in clear English, however in certain sections it needs to be rewritten substantially to meet the standard

of Commschem. In these sections, the language, expressions or grammar used are inadequate and negatively impact comprehensibility, up to the point where certain statements are rendered unintelligible; especially the section named ‘Selective hydrogenation of alkynols’, p16ff, requires significantly more work before it can be accepted. In general, the conclusions the authors have drawn are sound and the findings are novel, making this investigation an interesting read for a wider catalysis, organic chemistry and chemical engineering audience. Because the high demand for a highly selective, and ideally Pb-free, catalyst for these semi-hydrogenations, and the detailed characterisation of the new PdZn catalyst the authors have undertaken, I can recommend for this manuscript to be published in Commschem, after some changes outlined below and rewriting of the sections flagged above.

This is a list of changes that should be addressed before resubmitting a revised version:

1) Page 6, line 19: “Moreover, the adsorption energies of cis-BED on the PdZn (111) and Pd (111) are -0.27 eV and -0.55 eV, respectively...” – what about the adsorption energies of trans-BED? In the experimental hydrogenation section the authors have found that the alkene predominantly produced is cis not trans, which agrees with most literature. Nevertheless, information on the adsorption energies of trans-BED might be useful here to complete the picture.

2) Page 8, line 11: The term ZnO-T should be introduced here. This is done later on Page 26: "The obtained catalysts were assigned as Pd/ZnO-T, where T referred to the

reduction temperatures. " - this should be done here.

3) Page 9, figure 2: The diagram 2d is not clear. How do the differently coloured images of the catalyst correlate to the graph? The link needs to be explained.

4) Page 10, line 19: “Ultimately, in the reduction process, intermetallic PdZn NPs are formed through two-step processes including the reduction of PdO NPs as well as ZnO support partially and subsequently the diffusion of Zn atoms into Pd lattices”, What proof is there for migration of Zn into the Pd lattice?

5) Page 12, figure 3: It is not clear from the marking on the TEM images how distance measurements were conducted. What's the double dashed line on a), b) and c)?

6) Page 13, line 20: “The catalytic performance of SEA-prepared Pd/ZnO-T catalysts in 5 wt.% BYD aqueous system was evaluated at 2 MPa H₂, 80 °C, and contact time 29 gcat·h·mol⁻¹.” How was the contact time derived?

7) Page 15, figure 4: “(a) Conversion of BYD and product selectivity over Pd-based catalysts (Reaction conditions: 2 MPa H₂, 80 °C, and contact time 29 gcat. h mol⁻¹).”
– The bar graphs in figure 4a) show performance of catalysts prepared under different reduction temperatures; this is not made clear in the caption and also higher up in the text (end of page 13) and therefore the description should be altered to avoid confusion.

8) Page 16ff: “Significantly, the selectivity to cis-BED over Pd/ZnO-400 catalysts maintains a relatively stabilization with higher than 90.0% with increasing contact

time (Figure 4(c)). The SMSI effect with respect to Pd-Zn covalent interaction partially blocked the non-selective active sites of Pd NPs enhances the adsorption of the BYD with respect to that of the cis-BED, thus suppressing the over-hydrogenation of BED to product of BDO.”, this section and following sections need to be rewritten to make it clear what the authors would like to say.

9) Page 17, line 11: “This is the first report about the long-term stability of Pd-based catalyst in a continuous-flow semi-hydrogenation of BYD to cis-BED.” This is not the first continuous flow study of Pd-based catalysts for the semi-hydrogenation of BYD to cis-BED! Very substantial work has been carried out over the past years on the semi-hydrogenation of this particular molecule, 2-butyne-1,4-diol. The authors should reference recent work on this reaction, especially in industry relevant continuous flow hydrogenation settings, such as the work from Moreno-Marrodan et al (Beilstein J Org Chem. 2017; 13: 734–754, doi: 10.3762/bjoc.13.73), Tanielyan et al (Org. Process Res. Dev. 2017, 21, 3, 327–335, <https://doi.org/10.1021/acs.oprd.6b00375>) and Kundra et al (Chem. Eng. Process. 2020, 154, 108018, <https://doi.org/10.1016/j.cep.2020.108018>).

10) Page 17, line 22: “weightlessness peak” – What’s a weightlessness peak, can the authors explain?

11) Page 20, section named ”In situ DRIFT spectroscopy of the hydrogenation of propargyl alcohol” – this section is very lengthy and parts of it should be moved to the supporting information.

12) Page 28, line 8: "...and H₂-to-liquid feed ratio of 600." – Is this feed ratio based on volume/volume or mass/mass?

Response to the Reviewers' comments

Reviewer #1:

1. As shown in Figure 2, authors should show details for XPS measurements.

Were the tests performed at in-situ conditions or not?

Reply: Thank you very much for your comments. In our experiments, the XPS measurements were performed on Thermo VG Scientific ESCALAB250 electron spectrometers using a monochromatic Al Ka X-ray source. The C1s peak at 284.6 eV was used as standard for the calibration of binding energy. Due to the limitation of experimental conditions, the tests performed at non in-situ conditions. However, all Pd/ZnO-*T* samples were pre-reduced at different temperatures (150 °C, 200 °C, 300 °C, 400 °C, and 500 °C) before test and then transferred into the chamber by the Schlenk technology.

Modification: In section named 'Characterization of catalysts', the details for XPS measurements have been presented.

All Pd/ZnO-*T* samples were pre-reduced at different temperatures before test and then transferred into the chamber by the Schlenk technology while the spent Pd/ZnO-400 catalyst was directly moved into the chamber for the characterization.

2. Authors should show more details for the reaction mechanism, and I suggest to draw a proposed scheme.

Reply: Thank you very much for your suggestion. In our work, the mechanism of the

selective hydrogenation of BYD to *cis*-BED has been investigated basing on the *in situ* DRIFT spectroscopy and DFT calculations. Due to the isolated Pd active sites and modification their electronic structure by doping the Zn atoms, the BYD hydrogenation and *cis*-BED desorption are energetically favorable over the intermetallic PdZn particles. The over-hydrogenation and *cis-trans* isomeric reaction of BED over intermetallic PdZn have been extremely inhibited. Therefore, high selectivity to *cis*-BED with complete conversion of BYD has been achieved in the continuous-flow hydrogenation of BYD over intermetallic PdZn particles. As your suggestion, the reaction mechanism has been proposed, as shown in Scheme 1.

Modification: This theoretical simulation result nicely interprets the experimental results as shown in Figure 3 and the reaction mechanism of the selective hydrogenation of BYD over intermetallic PdZn was proposed, as shown in Scheme 1.

Scheme 1 Reaction mechanism of the selective hydrogenation of BYD over intermetallic PdZn.

3. Authors should give the Pd particle size distribution. Are there any smaller Pd particle size?

Reply: Thank you very much for your suggestion. The Pd particle size distribution has been detected by TEM. As shown in Figure 2(a), it can be found that the particles with average size of 7.0 nm were well-dispersed on the ZnO support.

Modification: The morphology and topography of the as-prepared Pd/ZnO-400 sample was revealed by TEM (Figure 2(a)). It is suggested that particles with average size 7.0 nm were well-dispersed with a narrow size distribution.

Figure 2. Morphologies of samples. (a) TEM image of Pd/ZnO-400 sample with inseting the particle size distribution. (b-c) HAADF-STEM images of Pd/ZnO-400 sample with inseting the corresponding model of catalyst and an atomic model of a unit cell of PdZn. (d) Intensity profile along the arrow in the HAADF-STEM image in (c) and the higher magnification HAADF-STEM image of PdZn particles, displaying characteristic ordering of Pd (navy blue) and Zn (sky blue) atoms. (e) A HAADF-STEM image of Pd/ZnO-400 sample along with the corresponding EDS elemental maps of Pd, Zn, and O.

Reviewer #2:

1) The manuscript is difficult to read and confusing. The lack of schemes of reaction, tables in the text, an overuse of acronyms (BID, BED, SMSI, RMSI IMC, SEA, DOS etc), an excessive reference to schemes in SI, make it hard to read. I suggest you to modify these points and to improve the English.

Reply: Thank you very much for your comments. For better understanding the selective hydrogenation of alkynols, the scheme of reaction has been moved from Scheme S1 into the Figure 3(a). The results of the semi-hydrogenation of other industrially relevant alkynols over Pd/ZnO-400 catalyst has been listed in Table 1, which has been supplied in the revised manuscript. In addition, some acronyms in the manuscript have been replaced by the full name, Such as (RMSI: reactive metal-support interaction; DOS: density of states). According to your suggestion, the expression in English has been improved with the assistance of Dr. Chi-Wing Tsang, Technological and Higher Education Institute of Hong Kong.

Modification:

- The scheme of reaction has been added in to Figure 3(a).

Figure 3. Selective hydrogenation properties of catalysts. (a) Reaction pathway for the hydrogenation of BYD. (b) Conversion of BYD and product selectivity over Pd/ZnO-*T* catalysts and the commercial Pd-based catalysts (Reaction conditions: 2 MPa H₂, 80 °C, and contact time 29 g_{cat.} h mol⁻¹). (c-d) Product distribution and selectivity of the BYD hydrogenation versus the contact time over Pd/ZnO-400 catalyst. (e) The stability testing of Pd/ZnO-400 catalyst in the BYD hydrogenation, in which the catalyst is regenerated at 110 h. (f) XPS spectra of Pd 3d region of the fresh and spent Pd/ZnO-400 catalyst.

Table 1 The semi-hydrogenation of other industrially relevant alkynols over Pd/ZnO-400 catalyst

Entry	Substrate	Product	Solvent	T (°C)	P (MPa)	τ^a (g _{cat.} ·h·mol ⁻¹)	Conv. (%)	Sel. (%)	TOFs ^b (min ⁻¹)
1			water	80	2	29	95.8	92.6	71
2			methanol	100	3	67	88.5	90.0	110
3			methanol	130	3	41	97.7	97.2	157
4			water	50	1	29	98.5	95.8	397

^a τ means the contact time.

^b The TOFs is calculated with the initial conversion of substrate <30%.

➤ The expression in English, such as:

The selective hydrogenation of alkynols using Pd-based IMCs catalysts have brought about widespread attention [18-23].

Therefore, to achieve highly efficient support IMCs catalysts, it was certainly worth designing an oxide-supported metal processor that highly dispersed metal nanoparticles (NPs) with firm anchorage on the support.

Additionally, there was no unequivocal conclusion about the XPS spectra of Zn 2p_{3/2} region (as shown in Figure S7) due to the strong signal from ZnO support.

Under the same reaction conditions, 100% conversion of BYD can be achieved over monometallic Pd-based catalysts, but the product distributions are relatively complicated.

Significantly, the selectivity to *cis*-BED over Pd/ZnO-400 catalysts maintained a relatively stable production with higher than 90.0% as contact time increased (Figure 3(d)).

Significantly, the selectivity to these enols presented quite stably with increasing the

contact time.

2) At page 5, the author introduces the catalyst PdZn/ZnO-400 but he doesn't explain the meaning of "400". Only after we can deduce it's the "reduction temperature".

Reply: Thank you very much for pointing it. The physical meaning of *T* should be assigned at first mention. In the revised manuscript, we have introduced it at page 8 and removed it in section of Methods.

Modification: "the optimized PdZn/ZnO-400 catalyst" has been changed into "the optimized PdZn/ZnO catalyst".

As shown in Figure S2, XRD patterns of ZnO and Pd/ZnO-*T* samples (*T* referred to the reduction temperatures) only presented the typical diffraction peaks of ZnO, indicating that the well-dispersed Pd NPs on the ZnO support are too small to be detected.

3) The synthesis of such catalyst was explained in part at the beginning (page 7-8, with reference to Figure S1 in SI, and then again at the end (page 25) with other details. In my opinion It's not so clear. Please explain better all details for the synthesis at the beginning or at the end.

Reply: Thank you very much for your suggestion. For better understanding the synthesis of Pd/ZnO catalysts by SEA method, all synthesis process analysis has been presented at the beginning with reference to Figure S1 in Supporting Information, and the synthesis method has shown in the experimental section.

Modification:

➤ At the beginning

Generally, the chemical environment, especial for the pH of solution, has a tremendous effect on the metal ion adsorbed on metal oxide support. Figure S1 showed that the point of zero charge of ZnO (specific surface area = $18 \text{ m}^2 \text{ g}^{-1}$) is ca. 8.2 and Na_2PdCl_4 as the Pd precursor could be achieved the maximum adsorption density under the optimal pH_{Final} of ca. 5.9-6.3. As a consequence, high dispersion can be achieved in Pd/ZnO catalyst under the effect of high ionic strength.

➤ At the end

Preparation of Pd/ZnO-*T* catalysts by SEA method. The strategy of SEA was employed for achieving the high dispersion of Pd NPs on ZnO support. Briefly, the functional groups on the ZnO surface could be protonated or deprotonated by adjusting the pH value to form a strongly charged surface, thereby anchoring oppositely charged metal ions. The point of zero charge of ZnO support was detected firstly and Na_2PdCl_4 as the Pd precursor was chose to adsorbed under acidic conditions (as described in the Supporting Information (Section S2)). To determine the pH at which the strongest surface interaction occurs, metal salt uptakes at various pH values were measured by ICP-AES, using a surface loading of $1000 \text{ m}^2 \text{ L}^{-1}$. After electrostatic adsorption, the sample were filtered, dried at $100 \text{ }^\circ\text{C}$ overnight, and then reduced in H_2 with a steady flow of 20 sccm for 2 h at various temperatures ($150 \text{ }^\circ\text{C}$, $200 \text{ }^\circ\text{C}$, $300 \text{ }^\circ\text{C}$, $400 \text{ }^\circ\text{C}$, and $500 \text{ }^\circ\text{C}$) with a heating rate of $5 \text{ }^\circ\text{C min}^{-1}$.

4) The article offer an accurate characterization of catalysts but it's extremely lacking in term of data of catalysis (e.g. experiments of catalysis, parameters to evaluate the productivity of catalysts, an accurate description of the flow reactor,

comparison with commercial catalysts).

Reply: Thank you very much for your comments. The related information of the liquid-phase hydrogenation of alkynols has been supplied in the revised manuscript, including the instrument, the parameters (reaction temperature, H₂ pressure, contact time, the amount of catalyst, and the concentration of liquid reactant). Additionally, the hydrogenation of 2-butyne-1,4-diol over commercial catalysts as comparison has been described in the experimental section. The details are as the follow responses.

• The authors use a flow reactor but they doesn't describe the instrument (Volume, length, materials, connections; check controls, process schemes) and the advantages respect to a batch reactor neither in the text nor in SI. They didn't report comparisons in batch conditions, please insert.

Reply: The description of the continuous flow reactor system has been supplied in the Methods. As shown in Scheme S1, the continuous flow reactor system is composed of a stainless steel pipe with 750 mm length and 8 mm internal diameter. In addition, we compare the catalytic performance of the selective hydrogenation of BYD over as-prepared Pd/ZnO-400 catalyst in the continuous flow reactor and the batch reactor. As shown in Figure S9, the as-prepared Pd/ZnO-400 catalyst also presented highly efficient in the selectivity hydrogenation of BYD to *cis*-BED in a batch reactor. High selectivity to *cis*-BED (97.3%) at the conversion of 99.5% can be achieved at 2.0 MPa H₂ and 80 °C for 2.0 h. However, with extending the reaction time to 5.0 h, the over-hydrogenation of *cis*-BED was carried out. The selectivity of *cis*-BED slightly decreased to 94.1% while the selectivity to BDO reached to 4.8%. Compared with the

selectivity hydrogenation of BYD in a batch reactor, the products obtained from the reaction in the continuous flow reactor was high stability at the fixed reaction conditions. In addition, it was adopted for industrial scale-up through the selectivity hydrogenation of BYD in the continuous flow reactor.

Modification:

➤ In the revised manuscript

As a comparison, the hydrogenation of BYD over Pd/ZnO-400 catalyst was also evaluated in a batch reactor. As shown in Figure S9, the Pd/ZnO-400 catalyst also presented high efficiency in the selectivity hydrogenation of BYD to *cis*-BED. However, the products obtained from the reaction in the continuous flow reactor had a higher stability at the fixed reaction conditions than in the batch reactor. In addition, it is usually adopted for industrial scale-up through the selectivity hydrogenation of BYD in the continuous flow reactor.

The liquid-phase selective hydrogenation of alkynols over Pd/ZnO-*T* samples, Pd/Al₂O₃, and the commercial Pd-based catalysts (Pd/C, Pd/TiO₂, Pd/SiO₂, Pd/CaCO₃, PdAg/Al₂O₃, Lindlar) were performed in a continuous flow reactor system consists of a stainless steel pipe (length, 750 mm; internal diameter, 8 mm) (as shown in Scheme S1). As comparison, the BYD hydrogenation performance over Pd/ZnO-400 catalyst in the batch reactor was also carried out (The details are described in SI).

➤ In the supporting information

4. Hydrogenation of BYD in the batch reactor.

Liquid-phase hydrogenation of BYD was carried out in a 50 mL stainless autoclave containing 0.1 g as-prepared Pd/ZnO catalyst, 25 mL of feedstock with 5 wt.% BYD, 1 wt.% 1,2-propylene glycol (as internal standard) and 96% water (as solvent) and 2.0 MPa H₂ at 80 °C. The reaction system was stirred vigorously (700 rpm) to eliminate the diffusion effect. Liquid aliquots were taken periodically using a dip tube. The products were analyzed by gas chromatography (Agilent 7890B-GC).

Scheme S1 The unit of high pressure fixed bed continuous flow stainless steel catalytic reactor.

Figure S9 (a) The variation of the relative concentration and (b) the products selectivities with reaction time for the hydrogenation of BYD over Pd/ZnO-400 catalyst in a batch reactor versus time (Reaction conditions: 0.1 g catalyst, reaction temperature 80 °C, 2 MPa H₂, and 25 g 5 wt.% BYD aqueous solution).

• **Tables with complete data of conversions, selectivity, STY, TOF, TON, productivity, should be added for hydrogenation of 2-butyn-1,4-diol but also for 2-methyl-3-butyn-2-ol, 3-hexynol, 3-phenyl-2-propyn-1-ol, described at page 18 only with TOFs.**

Reply: The Table with complete data about the results of the semi-hydrogenation of industrially relevant alkynols over Pd/ZnO-400 catalyst has been added in the revised manuscript.

Modification:

As presented in Table 1, the TOFs for the hydrogenation of these alkynols over Pt/ZnO-400 catalyst was follows: 2-methyl-3-butyn-2-ol (397 min⁻¹) > 3-hexyn-1-ol (157 min⁻¹) > 3-phenyl-2-propyn-1-ol (110 min⁻¹) > BYD (71 min⁻¹).

Table 1 The semi-hydrogenation of industrially relevant alkynols over Pd/ZnO-400 catalyst

Entry	Substrate	Product	Solvent	T (°C)	P (MPa)	τ^a (g _{cat.} ·h·mol ⁻¹)	Conv. (%)	Sel. (%)	TOFs ^b (min ⁻¹)
1			water	80	2	29	95.8	92.6	71
2			methanol	100	3	67	88.5	90.0	110
3			methanol	130	3	41	97.7	97.2	157
4			water	50	1	29	98.5	95.8	397

^a τ meant the contact time.

^b The TOFs was calculated with the initial conversion of substrate <30%.

• **The authors report only the contact time but it's not clear if they change only**

the flow of substrate, maintaining the content of catalyst (0.100 g ???) or both. In a flow process It's very important to show how productivity changes on the basis of parameters such as Hydrogen-flow, substrate-flow, time-contact, etc. Please specify.

Reply: The contact time is defined as the ratio of catalyst mass to molar flow rate of reaction substrate. In our experiment, the contact time has been changed by tuning the flow of substrate with maintaining the content of catalyst (0.1 g). For understanding the variation of productivity on the basis of parameters (reaction temperature, H₂ pressure, and the ratio of H₂/substrate), the related experimental data has been added.

Modification: The concept of contact time has been added in the experimental section of revised manuscript.

$$\text{Contact time (g}_{\text{cat.}} \text{ h mol}^{-1}) = m_{\text{cat.}} / (F * 60) \quad (1)$$

where $m_{\text{cat.}}$ (g) was the mass of catalyst (0.1 g), F (mol min⁻¹) was the molar flow rate of reactant.

• TOFs values obtained are good an in line with literature data but using hard reaction conditions in term of Pressure (2MPa), temperature (80°C), H₂/substrate ratio=600.

Reply: We investigate the selective hydrogenation of BYD over Pd/ZnO catalyst at mild conditions. As shown in Table R1, the TOF value indeed increases with increasing the reaction temperature and H₂ pressure. The hard reaction conditions in term of pressure and temperature promote the BYD adsorption/activation on the surface of Pd/ZnO-400 catalyst, improving the TOF value to some extent. In our

future research, the high TOFs for the hydrogenation of alkynols at mild conditions will be pursued.

Table R1 The effect of reaction conditions on the TOF values.

sample	T (°C)	P (MPa)	H ₂ /substrate ratio	TOF ^a (min ⁻¹)
Pd/ZnO-400	80	2	600	71
Pd/ZnO-400	80	0.1	600	56
Pd/ZnO-400	RT	2	600	37

^a Apparent TOF was calculated as moles of converted BYD per mole of Pd per minute with the relative low conversion.

• I suggest you to add experiments at different temperatures (e.g. room temperature), low pressures (e.g. 1-2 bar) and low H₂/Substrate ratio, changing the hydrogen flow. This type of reactions doesn't need hard conditions. Mild conditions favour selectivity.

Reply: According your suggestion, we added the experiments to explore the effects of reaction temperatures, H₂ pressures, and H₂/substrate ratios on the catalytic performances in the selectivity hydrogenation of BYD over Pd/ZnO-400 catalyst. Indeed, the selective hydrogenation of BYD to *cis*-BED is suitable under mild conditions. High selectivity to *cis*-BED (>97%) at almost complete conversion of BYD can be achieved at 80 °C, 0.5 MPa, and 100 H₂/substrate ratio.

Modification:

➤ In the revised manuscript

The effects of reaction temperatures, H₂ pressures, and the H₂/substrate volume ratios on the catalytic performances in the selectivity hydrogenation of BYD over

Pd/ZnO-400 catalyst in the continuous flow reactor were also explored, and the results were shown in Figure S10-S12. The selective hydrogenation of BYD to *cis*-BED was suitable under mild conditions. High selectivity to *cis*-BED (>97%) at almost complete conversion of BYD can be achieved at 80 °C, 0.5 MPa, and 100 H₂/substrate ratio. Although the conversion of BYD was boosted under the hard conditions, it easily led to the formation of by-products and lowering the selectivity to intermediate *cis*-BED. Figure S10 showed that *cis*-BED was prone to isomerization and over-hydrogenation at high temperature, while the over-hydrogenation of C=C bond to C-C bond and the cleavage of C-OH bond were enhanced under high H₂ pressure (as shown in Figure S11). Additionally, the selective hydrogenation of BYD to *cis*-BED in the continuous flow reactor can be operated at the relative low H₂/substrate volume ratio with 100 from the perspective of hydrogen economics (Figure S12) and it may be necessary to recycle hydrogen in the industrial scale-up. The effects of reaction temperatures (RT-100 °C), H₂ pressures (0.1-2.0 MPa), the volume ratio of H₂-to-liquid feed (50-600), and the contact time (0-87 g_{cat.}·h·mol⁻¹) on the catalytic performances in the selectivity hydrogenation of BYD over Pd/ZnO-400 catalyst were explored.

➤ In the supporting information

Figure S10 The effect of reaction temperature on the selective hydrogenation of BYD over Pd/ZnO-400 catalyst (Reaction conditions: 0.1 g catalyst, 2 MPa H₂, contact time 29 g_{cat.}·h·mol⁻¹, and the volume ratio of H₂ to liquid feed 600).

Figure S11 The effect of H₂ pressure on the selective hydrogenation of BYD over Pd/ZnO-400 catalyst (Reaction conditions: 0.1 g catalyst, reaction temperature 80 °C, contact time 29 g_{cat.}·h·mol⁻¹, and the volume ratio of H₂ to liquid feed 600).

Figure S12 The effect of the volume ratio of H₂ to liquid feed on the selective hydrogenation of BYD over Pd/ZnO-400 catalyst (Reaction conditions: 0.1 g catalyst, reaction temperature 80 °C, 2 MPa H₂, and contact time 29 g_{cat.}·h·mol⁻¹).

• **Authors compare their catalyst only with Lindlar catalyst and Pd/Al₂O₃. Please add comparison with other commercial catalysts (e.g. Pd/C; Pd/TiO₂; Pd/SiO₂; Pd/CaCO₃) in the same reaction conditions.**

Reply: Thank you very much for your suggestion. Some other commercial Pd-based catalysts including Pd/C, Pd/TiO₂, Pd/SiO₂, and Pd/CaCO₃ have been evaluated in the selective hydrogenation of BYD. The conversion of BYD and the products distribution are displayed in Figure 3(b). Under the same reaction conditions, the commercial monometallic Pd-based catalysts present high activity and achieve 100% conversion of BYD while the product distributions are relative complex with relative low selectivity to *cis*-BED. The over-hydrogenation and *cis-trans* BED isomerization is prone to carry out. It further confirms that the Pd/ZnO-400 catalyst contained PdZn as active sites presents a promising application prospects in the selectivity hydrogenation of BYD to *cis*-BED.

Modification: Figure 3(b) has been changed and the related discussion has been added in the revised manuscript.

Under the same reaction conditions, 100% conversion of BYD can be achieved over monometallic Pd-based catalysts, but the product distributions are relatively complicated. The selectivity to *cis*-BED was only 7.5% for Pd/Al₂O₃, 1.5% for Pd/SiO₂, 0.4% for Pd/C, 19.3% for Pd/TiO₂, and 94.3% for Pd/CaCO₃, respectively. The over-hydrogenation of *cis*-BED was prone to carry out over monometallic Pd-based catalysts, especially for the Pd/SiO₂ and Pd/C catalysts with weak metal-support interaction. In addition, the hydrogen-mediated *cis-trans* BED isomerization was observed on Pd/TiO₂ catalyst and the selectivity to *trans*-BED reaches 28.9%. This phenomenon was similar with the selective stereochemical catalysis over RhSb/SiO₂ [41].

The liquid-phase selective hydrogenation of alkynols over Pd/ZnO-*T* samples, Pd/Al₂O₃, and the commercial Pd-based catalysts (Pd/C, Pd/TiO₂, Pd/SiO₂, Pd/CaCO₃, PdAg/Al₂O₃, Lindlar) were performed in a continuous flow reactor system consists of a stainless steel pipe (length, 750 mm; internal diameter, 8 mm) (as shown in Scheme S1).

Figure 3. Selective hydrogenation properties of catalysts. (a) Reaction pathway for the hydrogenation of BYD. (b) Conversion of BYD and product selectivity over Pd/ZnO-*T* catalysts and the commercial Pd-based catalysts (Reaction conditions: 2 MPa H₂, 80 °C, and contact time 29 g_{cat.} h mol⁻¹). (c-d) Product distribution and selectivity of the BYD hydrogenation versus the contact time over Pd/ZnO-400 catalyst. (e) The stability testing of Pd/ZnO-400 catalyst in the BYD hydrogenation, in which the catalyst is regenerated at 110 h. (f) XPS spectra of Pd 3d region of the fresh and spent Pd/ZnO-400 catalyst.

[41]Furukawa, S., Ochi, K., Luo, H., Miyazaki, M. & Komatsu, T. Selective stereochemical catalysis controlled by specific atomic arrangement of ordered alloys. *ChemCatChem* **7**, 3472–3479, (2015).

• **Please describe better in situ regeneration by H₂, reaction conditions and characterization of catalyst fresh, used and regenerated.**

Reply: In the stability testing, the Pd/ZnO-400 catalyst was *in situ* regeneration by reducing it under 48 sccm hydrogen flow at 400 °C for 2 h before terminating the liquid feedstock. After in situ regeneration, the second stage stability testing was carried out at the same reaction conditions (2 MPa H₂, 80 °C, and contact time 29 g_{cat}. h mol⁻¹). The spent Pd/ZnO-400 catalyst was characterized by TG and XPS for understanding its structure.

Modification: After terminating the liquid feedstock and *in situ* reducing the catalyst under 48 sccm hydrogen flow at 400 °C for 2 h, the second stage stability test was carried out at the same reaction conditions.

TG/DTG analysis of the PdO/ZnO sample and the spent Pd/ZnO-400 catalyst was performed on a Mettler-Toledo TGA/SDTA3+ jointing with Pfeiffer OmniStar mass spectrometer.

The XPS measurements were performed on Thermo VG Scientific ESCALAB250 electron spectrometers using a monochromatic Al K α X-ray source. All Pd/ZnO-*T* samples were pre-reduced at different temperatures before test and then transferred into the chamber by the Schlenk technology while the spent Pd/ZnO-400 catalyst was directly moved into the chamber for the characterization.

• **No information about leaching in solution was considered. Please measure metal leaching in solution by ICP.**

Reply: According to your suggestion, the content of metals in the reaction solution has been detected by ICP-AES. There is no leaching of Pd and Zn in solution, which may be below the detection limit. The related description has been added in the revised manuscript.

Modification: Through analyzing the content of metals in the reaction solution by ICP-AES, no leaching of Pd and Zn was observed (below the detection limit).

The practical loading of Pd in the samples and the content of Pd and Zn in the reaction solution were analyzed by ICP-AES.

5) the article doesn't offer strong element of novelty both in term of materials and catalysis, already described in literature and not sufficiently cited (e.g. among them: for materials: ACS catal 2017,7,2, 1491; Catal. Today 2018, 330, 2; J. Phys. Chem C 2011,115, 8457; ACS Appl. Nanomaterials 2019, 2,5,3307; for Hydrogenation of alkynes: Chem Eng Proc. Proc Intensification 2020, 154, 108018; ACS Sust Chem Eng 2019, 7, 13, 11050; J. Catal, 2014, 311, 212; Beilstein. J. Org Chem 2017, 13, 734; ChemCatChem 2017, 9, 3245). Please add some references for both arguments.

Reply: Thank you very much for your comments. Designing highly active and stable Pb-free Pd-based catalysts without introducing of surfactants and stabilizers is vital for large-scale and efficient production of *cis*-enols through continuous-flow hydrogenation of alkynols, which is a critical process in the fine and intermediate

chemical industry. In the past decade, some related studies on the continuous-flow processes for the catalytic partial hydrogenation of alkynols over Pd-based catalysts have emerged. The activity of the related catalysts as well as their stability and selectivity for target products were shown to be determined by the intrinsic metal active sites, promoters (second metals, organic ligands, and stabilizers), and the nature of supports (pore structure, acid- base properties, and SMSI). Compared with traditional Lindlar- type catalysts, such Pd- based intermetallics can be extensively used for the selective hydrogenation of alkynols, as such doping enhances selectivity for enols while avoiding the hazards associated with toxic additives. However, the related research is lack, although the Pd-based intermetallics catalysts for the semi-hydrogenation of acetylene are widely focused on and researched. In this work, the finely surface engineering technology in oxide-supported metal (particles) has been applied to controllable tune the geometric and electronic structure of Pd active sites for achieving the directional hydrogenation of alkynols. Though coupling strategy of strong electrostatic adsorption and reactive metal-support interaction, intermetallic PdZn/ZnO catalyst is designed. Driven by hydrogen at high temperature, the formation of intermetallic PdZn nanoparticles with electron-poor ($\text{Pd}^{\delta+}$) active sites on a ZnO supported Pd catalyst significantly boosts the thermodynamic selectivity with respect to the mechanistic selectivity and therefore enhances the selectivity to *cis*-enols. The optimized Pd/ZnO-400 catalyst exhibits superior activity, unique selectivity, and long term stability in the continuous-flow semi-hydrogenation of alkynols. Based on the *in situ* DRIFTS and simulation calculation, the

semi-hydrogenation of alkynols over intermetallic PdZn/ZnO has been proposed. Overall, on the basis of previous studies, this work is innovative in the design of intermetallic compound catalyst, the optimization of reaction process, and the understanding of reaction mechanism.

Modification: The related references have been added in the revised manuscript.

Owing to the large market demand for enols, continuous-flow semi-hydrogenation of alkynols over heterogeneous catalysts will be an inevitable trend to obtain high-quality products under large-scale applications in virtue of their process safety, environmental friendliness, and economic benefits [5-10].

Intermetallic compounds (IMCs) with ordered crystal structures and special electronic structure play a key role considering the practical manufacturing of heterogeneous catalysts [14-17].

The introduction of a second metal tunes the ensemble and ligand effects of metallic Pd significantly, enhancing the selective hydrogenation of alkynes [24-26].

[5] Liguori, F. & Barbaro, P. Green semi-hydrogenation of alkynes by Pd@borate monolith catalysts under continuous flow. *J. Catal.* **311**, 212–220 (2014).

[6] Tanielyan, S. K., More, S. R., Augustine, R. L. & Schmidt, S. R. Continuous liquid-phase hydrogenation of 1,4-butyne-1,3-diol to high-purity 1,4-butanediol over particulate Raney nickel catalyst in a fixed bed reactor. *Org. Process Res. Dev.* **21**, 327–335 (2017).

[7] Moreno-Marrodan, C., Liguori, F. & Barbaro, P. Continuous-flow processes for the catalytic partial hydrogenation reaction of alkynes. *Beilstein J. Org. Chem.* **13**,

734–754 (2017).

[8] Liguori, F. et al. Unconventional Pd@sulfonated silica monoliths catalysts for selective partial hydrogenation reactions under continuous flow. *ChemCatChem* **9**, 3245–3258 (2017).

[9] Fukazawa, A. et al. A new approach to stereoselective electrocatalytic semihydrogenation of alkynes to Z- alkenes using a proton-exchange membrane reactor. *ACS Sustainable Chem. Eng.* **7**, 11050–11055 (2019).

[10]Kundra, M. et al. Continuous flow semi-hydrogenation of alkynes using 3D printed catalytic static mixers. *Chem. Eng. Process.* **154**, 108018 (2020).

[16]Dasgupta, A. & Rioux, R. M. Intermetallics in catalysis: An exciting subset of multimetallic catalysts. *Catal. Today* **330**, 2–15 (2019).

[24]Tew, M. W., Emerich, H. & van Bokhoven, J. A. Formation and characterization of PdZn Alloy: A very selective catalyst for alkyne semihydrogenation. *J. Phys. Chem. C* **115**, 8457–8465 (2011).

[25]Pei, G. X., et al. Performance of Cu-Alloyed Pd single-atom catalyst for semihydrogenation of acetylene under simulated front-end conditions. *ACS Catal.* **7**, 1491–1500 (2017).

[26]Miyazaki, M., Furukawa, S., Takayama, T., Yamazoe, S. & Komatsu, T. Surface modification of PdZn nanoparticles via galvanic replacement for the selective hydrogenation of terminal alkynes. *ACS Appl. Nano Mater.* **2**, 3307–3314 (2019).

6) The suppression of overhydrogenation of BED to BDO, and the selectivity to cis-BED, cited at page 16, can be explained by DFT analysis at page 22 ??

Explain better. It's not clear. In case, please add in the text, at page 16, some reference to this part.

Reply: Indeed, the subsequent DFT analysis can explain the suppression the over-hydrogenation of BED to product of BDO over intermetallic PdZn particles. For better understanding this phenomenon, the related research work on the semi-hydrogenation of acetylene over ultra-small intermetallic PdZn particles has been cited in the revised manuscript.

Modification: The SMSI effect with respect to Pd-Zn covalent interaction partially blocked the non-selective active sites of Pd NPs enhances the adsorption of the BYD with respect to that of the *cis*-BED, thus suppressing the over-hydrogenation of BED to product of BDO. This was similar with that the energetically favorable path for acetylene hydrogenation and ethylene desorption over the ultra-small intermetallic PdZn particles [43].

[43]Hu, M. et al. MOF-confined sub-2 nm atomically ordered intermetallic PdZn nanoparticles as high-performance catalysts for selective hydrogenation of acetylene. *Adv. Mater.* **30**, 1801878 (2018).

7) Why do you make DRIFT spectroscopy with propargyl alcohol ? Why do you use such a danger /toxic compound, never used in previous cited catalytic experiments? Why don't you use 2-butyne-1,4-diol or others?

Reply: That is a good question. The previous researches on the hydrogenation of acrolein and α - amino ester probed by FTIR spectroscopy provide a great guide on the explore the hydrogenation of alkynol by applying in situ DRIFT spectroscopy

[1-3]. In our research, the *in situ* DRIFT spectroscopy with propargyl alcohol has been utilized for identifying the reactive surface intermediates in the selective hydrogenation of alkynol over PdZn/ZnO catalyst. Generally, both alkyne and C-OH group are presented in propargyl alcohol, which has a similar structure with 2-butyne-1,4-diol. It can be as a candidate to explore the mechanism of selective hydrogenation of C≡C bonds to C=C bonds and the possible hydrogenolysis or hydrodeoxygenation of C-OH bonds through DRIFT spectroscopy. In addition, the boiling point of propargyl alcohol (114.5 °C) is relative lower than that of 2-butyne-1,4-diol (228 °C). It is very convenient to feed into the Praying Manitis DRIFTS reactor under the action of Ar as carrier gas to detect the adsorption or hydrogenation over Pd/ZnO sample.

References:

1. Dostert, K.-H., O'Brien, C. P., Ivars-Barceló, F., Schauer mann, S. & Freund, H.-J. Spectators control selectivity in surface chemistry: Acrolein partial hydrogenation over Pd. *J. Am. Chem. Soc.* **137**, 13496–13502 (2017).
2. Dostert, K.-H. et al. Selective partial hydrogenation of acrolein on Pd: A mechanistic study. *ACS Catal.* **7**, 5523–5533 (2017).
3. Zhang, L., Lohrasbi, M., Tumuluri, U. & Chuang, S. S. C. Asymmetric hydrogenation of α - amino ester probed by FTIR spectroscopy. *Org. Process Res. Dev.* **20**, 1668–1676 (2016).

8) DFT calculations are reported at the beginning (page 5-7) and then at the end of the paper (page 22-23) with reference to figures and schemes at the beginning

and to SI. It makes difficult to read and to understand the reasoning. I suggest to write only a chapter about DFT calculation at the end of the paper to explain all details for thermodynamics and mechanistic selectivity. In general, for a better understanding the text, I suggest to follow a scheme: synthesis of catalyst, complete characterization of catalyst, catalytic experiments in flow and batch, DFT calculations and DRIFT spectroscopy to explain the selectivity and mechanism, conclusion, methods. Now everything it's mixed and confusing.

Reply: Thank you very much for your suggestion. For better understanding the text, the structure of the article has been re-adjusted. Now, we wrote the Results in the following order: PdZn/ZnO catalyst prepared by SEA and following reduction, Selective hydrogenation of alkynols, *In situ* DRIFT spectroscopy of the hydrogenation of propargyl alcohol, DFT analysis of reaction mechanism. In addition, the descriptions and the Figures has been reconstructed. Some Figures have been transferred in the supporting information.

Modification:

➤ In the revised manuscript

PdZn/ZnO catalyst prepared by SEA and following reduction. Site modification and isolation was known to be a powerful strategy of enhancing the selectivity of Pd-based catalysts for the semi-hydrogenation of alkynols to enols [32]. Here, a PdZn/ZnO catalyst have been designed by the relative coupling strategy of SEA and following high temperature reduction, in which Pd active sties were modified by Zn *via* formation of intermetallic PdZn, as shown in Figure 1(a) [33]. Due to the

strong covalent Pd–Zn interaction, the pronounced charge correlation regions around the Pd atom owing to electron accumulation can be observed in Figure 1(b). The calculated results showed that the Mulliken atomic charges are ca. -0.17 and 0.18 e for the superficial Pd and Zn atom, respectively. Based on the fact that the *d*-band width of metal directly affects its capacity to adsorb unsaturated chemical bonds by the semi-empirical method [34], the intermetallic PdZn/ZnO catalyst may present high potential in the selective hydrogenation of alkynols.

DFT analysis of reaction mechanism. To better understand the underlying mechanism, the calculated total electronic density of states of BYD, BED, Pd, and PdZn and the adsorption configurations and energies of reaction species over Pd (111) and PdZn (111) were calculated. As shown in Figure S16, the *d* bands center of intermetallic PdZn presented a negative deviation and the bandwidth was reduced significant in comparison with metallic Pd, which may be ascribed to the missing *d-d* interactions and lattice strain, leading to the dramatic change in the catalytic properties [50]. Interestingly, the *d* bands of PdZn overlapped with the *p* bands of BYD slightly but there was almost no overlay with that of BED. However, they overlapped and interweaved among the *d* bands of metallic Pd with the *P* bands of BYD and BED. From the point of electronic effect, it can deduce that intermetallic PdZn boosts the selectivity to BED and suppresses its over-hydrogenation comparing the metallic Pd due to the weaker interaction. Based on the thermodynamically stable and mostly exposed surfaces of PdZn (111) and Pd (111), the configurations and energies of the most stable BYD, *cis*-BED, and *trans*-BED adsorption on these

surfaces were calculated, as shown in Figure S17. The adsorption energies of BYD on the PdZn (111) and Pd (111) were -0.97 eV and -1.30 eV, respectively, emerging highly efficient reactivity of Pd sites of the PdZn (111) and Pd (111). Moreover, the adsorption energies of *cis*-BED on the PdZn (111) and Pd (111) were -0.27 eV and -0.55 eV, respectively, which represented that the desorption of *cis*-BED on the surface of PdZn had priority over that on Pd. The same results can also be found for the *trans*-BED while the *cis-trans* BED isomerization should be further explored. In view of the above-mentioned facts, it can deduce that the thermodynamic selectivity has been enhanced and the over-hydrogenation or hydrogenolysis of BED over intermetallic PdZn may be inhibited significantly. Regulating the Pd active sites to the completely isolated by Zn given rise to preferential π - adsorption of $C\equiv C$ bonds of BYD, and decreased the adsorption energies significantly, indicating the intermetallic PdZn is a promising catalyst for the semi-hydrogenation of BYD to BED.

Figure 1. Structure characterization of intermetallic PdZn. (a) Schematic illustrating PdZn/ZnO catalyst. (b) Deformation density of intermetallic PdZn. (c) TG/DTG

profiles for the Pd/ZnO sample under 5 vol.% H₂/Ar from RT to 520 °C with 5 °C min⁻¹. (d) H₂-TPR profiles of Pd/ZnO and ZnO samples in 10 vol.% H₂/Ar up to 500 °C with 5 °C min⁻¹. (e) XPS spectra of Pd 3d region of unreduced PdO/ZnO and Pd/ZnO-*T* samples.

➤ In the revised manuscript

Figure S6. The molar ratio of PdZn/Pd versus the reduction temperature (PdO: navy blue, Pd: wine red, PdZn: yellow).

Figure S16 The calculated total electronic density of states of BYD, BED, Pd, and PdZn.

Figure S17 Adsorption configurations and energies of BYD, *cis*-BED, and *trans*-BED on PdZn (111) and Pd (111), respectively.

Reviewer #3:

The manuscript is structured logically and written in most parts in clear English, however in certain sections it needs to be rewritten substantially to meet the standard of Commschem. In these sections, the language, expressions or grammar used are inadequate and negatively impact comprehensibility, up to the point where certain statements are rendered unintelligible; especially the section named ‘Selective hydrogenation of alkynols’, p16ff, requires significantly more work before it can be accepted.

Reply: Thank you very much for your comments. According to your suggestion, the expression in English has been improved with the assistance of Dr. Chi-Wing Tsang, Technological and Higher Education Institute of Hong Kong.

Modification: The language, expressions and grammar used in the manuscript have been modified, such as,

However, at high reduction temperatures, the reconstruction of catalyst as well as the formation of intermetallic PdZn continued to occupy center stage, which enhanced the ensemble and ligand effects on the Pd active sites, boosting the semi-hydrogenation of BYD to *cis*-BED.

It clearly showed that the selectivity to *cis*-BED can reach to ca. 92% with the BYD conversion of 96% over the as-prepared Pd/ZnO-400 catalyst.

The TOF value of Pd/ZnO-400 catalyst with the PdZn/Pd molar ratio 1:1 (71 min^{-1}) was slightly lower than that of Pd/ZnO-150 catalyst with pure Pd NPs supported on ZnO (90 min^{-1}), which further indicated that partial active sites were converged. Under the same

reaction conditions, 100% conversion of BYD can be achieved over monometallic Pd-based catalysts, but the product distributions are relatively complicated.

Significantly, the selectivity to *cis*-BED over Pd/ZnO-400 catalysts maintained a relatively stable production with higher than 90.0% as contact time increased (Figure 3(d)). The SMSI effect with respect to Pd-Zn covalent interaction partially blocked the non-selective active sites of Pd NPs preferred the thermodynamic selectivity comparing to the mechanistic selectivity, thus suppressing the over-hydrogenation of *cis*-BED to product of BDO. This was similar with the observation on the energetically favorable path for acetylene hydrogenation and ethylene desorption over the ultra-small intermetallic PdZn particles [43].

It can be deduced that the non-selective PdH_x active sites with high activity in the surface of Pd/ZnO-400 catalyst were buried in oblivion as the extension of the reaction time.

In situ regeneration by H₂ reduction treatment is a powerful strategy to eliminate the strongly adsorbed species on the active sites and strengthen the interaction between inter(metallic) active center with oxides support [44].

1) Page 6, line 19: “Moreover, the adsorption energies of *cis*-BED on the PdZn (111) and Pd (111) are -0.27 eV and -0.55 eV, respectively...” – what about the adsorption energies of *trans*-BED? In the experimental hydrogenation section the authors have found that the alkene predominantly produced is *cis* not *trans*, which agrees with most literature. Nevertheless, information on the adsorption energies of *trans*-BED might be useful here to complete the picture.

Reply: Thank you very much for your suggestion. The adsorption energies of *trans*-BED on the PdZn (111) and Pd (111), respectively, have been calculated. As shown in Figure S18, the adsorption energies of *trans*-BED on the PdZn (111) and Pd (111) are -0.27 eV and -0.61 eV, respectively, which are almost similar with the adsorption energies of *cis*-BED on the PdZn (111) and Pd (111). Just considering the adsorption energy, it is hardly to clarify that the selective hydrogenation of BYD predominantly produced is *cis*-BED not *trans*-BED. However, the energy distribution diagrams of the hydrogenation pathways of C≡C bonds in BYD (Figure 5) present that the hydrogenation barriers of *trans*-TS_{2Pd} is close to that of *cis*-TS_{2Pd}, indicating that *cis/trans*-isomerism of BED can be obtained over metallic Pd catalyst. While the value of *cis*-TS_{2PdZn} is much lower than that of *trans*-TS_{2PdZn}, which indicates that there is extremely unlikely to carry out the *cis/trans*-isomerism of BED over intermetallic PdZn.

Modification:

➤ In the revised manuscript

Based on the thermodynamically stable and mostly exposed surfaces of PdZn (111) and Pd (111), the configurations and energies of the most stable BYD, *cis*-BED, and *trans*-BED adsorption on these surfaces were calculated, as shown in Figure S17. The adsorption energies of BYD on the PdZn (111) and Pd (111) were -0.97 eV and -1.30 eV, respectively, emerging highly efficient reactivity of Pd sites of the PdZn (111) and Pd (111). Moreover, the adsorption energies of *cis*-BED on the PdZn (111) and Pd (111) were -0.27 eV and -0.55 eV, respectively, which represented that the desorption

of *cis*-BED on the surface of PdZn had priority over that on Pd. The same results can also be found for the *trans*-BED while the *cis-trans* BED isomerization should be further explored. In view of the above-mentioned facts, it can deduce that the thermodynamic selectivity has been enhanced and the over-hydrogenation or hydrogenolysis of BED over intermetallic PdZn may be inhibited significantly.

➤ In the Supporting Information

The adsorption configurations and energies of BYD, *cis*-BED, and *trans*-BED on PdZn (111) and Pd (111), respectively, have been shown in Supporting information (Figure S17).

Figure S17 Adsorption configurations and energies of BYD, *cis*-BED, and *trans*-BED on PdZn (111) and Pd (111), respectively.

2) Page 8, line 11: The term ZnO-*T* should be introduced here. This is done later on Page 26:"The obtained catalysts were assigned as Pd/ZnO-*T*, where T

referred to the reduction temperatures. " - this should be done here.

Reply: Thank you very much for pointing this mistake. The physical meaning of T should be assigned at first mention. In the revised manuscript, we have introduced it at page 7 and removed it in section of Methods.

Modification:

As shown in Figure S2, XRD patterns of ZnO and Pd/ZnO- T samples (T referred to the reduction temperatures) only presented the typical diffraction peaks of ZnO, indicating that the well-dispersed Pd NPs on the ZnO support are too small to be detected.

3) Page 9, figure 2: The diagram 2d is not clear. How do the differently coloured images of the catalyst correlate to the graph? The link needs to be explained.

Reply: As show in Figure 2d (displayed in Figure S6 after revised), the molar ratio of PdZn/Pd calculated based on the XPS peak area (Figure 1e) has been related to the reduction temperature. In order to understand the configuration of the Pd/ZnO- T catalyst, the scheme of catalysts has been described. The PdO, Pd, and PdZn nanoparticles have been shown in navy blue, wine red, and yellow images, respectively.

Modification: In the supporting information, the mean of colored images (as shown in Figure S6) has been added.

Figure S6. The molar ratio of PdZn/Pd versus the reduction temperature (PdO: navy blue, Pd: wine red, PdZn: yellow).

4) Page 10, line 19: “Ultimately, in the reduction process, intermetallic PdZn NPs are formed through two-step processes including the reduction of PdO NPs as well as ZnO support partially and subsequently the diffusion of Zn atoms into Pd lattices”, What proof is there for migration of Zn into the Pd lattice?

Reply: From the TG/DTG-MS measurements of Pd/ZnO sample in H₂ atmosphere and H₂-TPR, we can only infer that the formation of PdZn intermetallic compound due to the SMSI effect. However, it provides the directly evidence to proof the formation of PdZn intermetallic compound for the Pd/ZnO-400 sample and the Zn in the Pd lattice by the XPS and HAADF-STEM measurements.

Modification: Ultimately, in the reduction process, intermetallic PdZn NPs are formed through two-step processes including the reduction of PdO NPs as well as

ZnO support partially and subsequently the diffusion of Zn atoms into Pd lattices due to the SMSI effect, which will be confirmed by subsequent X-ray photoelectron spectroscopy (XPS) and high-angle annular dark-field scanning transmission electron microscopy (HAADF-STEM) measurements.

5) Page 12, figure 3: It is not clear from the marking on the TEM images how distance measurements were conducted. What's the double dashed line on a), b) and c)?

Reply: The HAADF-STEM images of Pd/ZnO-400 sample have been analyzed by GMS digital micrograph software. Through the tool of Profile, We can get a rectangular box perpendicular to lattice stripe, as shown in Figure R1(a). In the Profile window (as shown in Figure R1(b) and (c)), a dotted line box can be dragged out and the distance is displayed. Finally, the crystal plane index of sample can be obtained by comparing the crystal plane spacing of XRD standard spectrum. In our manuscript, the double dashed lines on Figure 2(b-c) have been used to label the lattice stripes.

Figure R1 The analysis of HRTEM image of Pd/ZnO-400 sample by GMS digital

micrograph software.

6) Page 13, line 20: “The catalytic performance of SEA-prepared Pd/ZnO-T catalysts in 5 wt.% BYD aqueous system was evaluated at 2 MPa H₂, 80 °C, and contact time 29 g_{cat.}·h·mol⁻¹.” How was the contact time derived?

Reply: The contact time is defined as the ratio of catalyst mass to molar flow rate of reaction substrate.

Modification: The concept of contact time has been added in the experimental section of revised manuscript.

$$\text{Contact time (g}_{\text{cat.}} \text{ h mol}^{-1}) = m_{\text{cat.}} / (F * 60) \quad (1)$$

where $m_{\text{cat.}}$ (g) was the mass of catalyst (0.1 g), F (mol min⁻¹) was the molar flow rate of reactant.

7) Page 15, figure 4: “(a) Conversion of BYD and product selectivity over Pd-based catalysts (Reaction conditions: 2 MPa H₂, 80 °C, and contact time 29 g_{cat.} h mol⁻¹).” – The bar graphs in figure 4a) show performance of catalysts prepared under different reduction temperatures; this is not made clear in the caption and also higher up in the text (end of page 13) and therefore the description should be altered to avoid confusion.

Reply: Both the as-prepared Pd/ZnO-T catalysts and the commercial Pd-based catalysts have been evaluated in the selective hydrogenation of BYD under 2 MPa H₂, 80 °C, and contact time 29 g_{cat.} h mol⁻¹. For avoiding confusion, the caption of Figure 3b has been changed into “Conversion of BYD and product selectivity over Pd/ZnO-T catalysts and the commercial Pd-based catalysts (Reaction conditions: 2 MPa H₂, 80

°C, and contact time $29 \text{ g}_{\text{cat}} \cdot \text{h mol}^{-1}$)”

Modification: The caption of Figure 3b has been modified in the revised manuscript.

Figure 3. Selective hydrogenation properties of catalysts. (a) Reaction pathway for the hydrogenation of BYD. (b) Conversion of BYD and product selectivity over Pd/ZnO-*T* catalysts and the commercial Pd-based catalysts (Reaction conditions: 2 MPa H₂, 80 °C, and contact time $29 \text{ g}_{\text{cat}} \cdot \text{h mol}^{-1}$). (c-d) Product distribution and selectivity of the BYD hydrogenation versus the contact time over Pd/ZnO-400

catalyst. (e) The stability testing of Pd/ZnO-400 catalyst in the BYD hydrogenation, in which the catalyst is regenerated at 110 h. (f) XPS spectra of Pd 3d region of the fresh and spent Pd/ZnO-400 catalyst.

8) Page 16ff: “Significantly, the selectivity to *cis*-BED over Pd/ZnO-400 catalysts maintains a relatively stabilization with higher than 90.0% with increasing contact time (Figure 4(c)). The SMSI effect with respect to Pd-Zn covalent interaction partially blocked the non-selective active sites of Pd NPs enhances the adsorption of the BYD with respect to that of the *cis*-BED, thus suppressing the over-hydrogenation of BED to product of BDO.”, this section and following sections need to be rewritten to make it clear what the authors would like to say.

Reply: To void the confusion, these sections have been rewritten. The Pd/ZnO-400 catalyst with intermetallic PdZn active sites presents high efficient and high selectivity to *cis*-BED in the hydrogenation of BYD, which may be attributed that the formation of Pd-Zn covalent interaction partially blocks the non-selective active sites of Pd NPs, preferring to the thermodynamic selectivity comparing to the mechanistic selectivity. Therefore, the over-hydrogenation of *cis*-BED to BDO has been inhibited. The same phenomenon was also found in the semi-hydrogenation of acetylene over the ultra-small intermetallic PdZn particles. The related reference has been added in the revised manuscript.

Modification: The SMSI effect with respect to Pd-Zn covalent interaction partially blocked the non-selective active sites of Pd NPs preferred the thermodynamic selectivity comparing to the mechanistic selectivity, thus suppressing the over-hydrogenation of *cis*-BED to product of BDO. This was similar with the

observation on the energetically favorable path for acetylene hydrogenation and ethylene desorption over the ultra-small intermetallic PdZn particles [43].

[43]Hu, M. et al. MOF-confined sub-2 nm atomically ordered intermetallic PdZn nanoparticles as high-performance catalysts for selective hydrogenation of acetylene. *Adv. Mater.* **30**, 1801878 (2018).

9) Page 17, line 11: “This is the first report about the long-term stability of Pd-based catalyst in a continuous-flow semi-hydrogenation of BYD to cis-BED.” This is not the first continuous flow study of Pd-based catalysts for the semi-hydrogenation of BYD to cis-BED! Very substantial work has been carried out over the past years on the semi-hydrogenation of this particular molecule, 2-butyne-1,4-diol. The authors should reference recent work on this reaction, especially in industry relevant continuous flow hydrogenation settings, such as the work from Moreno-Marrodan et al (Beilstein J Org Chem. 2017; 13: 734–754, doi: 10.3762/bjoc.13.73), Tanielyan et al (Org. Process Res. Dev. 2017, 21, 3, 327–335, <https://doi.org/10.1021/acs.oprd.6b00375>) and Kundra et al (Chem. Eng. Process. 2020, 154, 108018, <https://doi.org/10.1016/j.cep.2020.108018>).

Reply: Thank you very much for your comments. The sentence “This is the first report about the long-term stability of Pd-based catalyst in a continuous-flow semi-hydrogenation of BYD to *cis*-BED.” has been removed. The recent works on the selective hydrogenation of BYD in continuous flow reactor have been supplied in the revised manuscript.

Modification: Owing to the large market demand for enols, continuous-flow

semi-hydrogenation of alkynols over heterogeneous catalysts will be an inevitable trend to obtain high-quality products under large-scale applications in virtue of their process safety, environmental friendliness, and economic benefits [5-10].

[5] Liguori, F. & Barbaro, P. Green semi-hydrogenation of alkynes by Pd@borate monolith catalysts under continuous flow. *J. Catal.* **311**, 212–220 (2014).

[6] Tanielyan, S. K., More, S. R., Augustine, R. L. & Schmidt, S. R. Continuous liquid-phase hydrogenation of 1,4-butyne-1,3-diol to high-purity 1,4-butanediol over particulate Raney nickel catalyst in a fixed bed reactor. *Org. Process Res. Dev.* **21**, 327–335 (2017).

[7] Moreno-Marrodan, C., Liguori, F. & Barbaro, P. Continuous-flow processes for the catalytic partial hydrogenation reaction of alkynes. *Beilstein J. Org. Chem.* **13**, 734–754 (2017).

[8] Liguori, F. et al. Unconventional Pd@sulfonated silica monoliths catalysts for selective partial hydrogenation reactions under continuous flow. *ChemCatChem* **9**, 3245–3258 (2017).

[9] Fukazawa, A. et al. A new approach to stereoselective electrocatalytic semihydrogenation of alkynes to Z-alkenes using a proton-exchange membrane reactor. *ACS Sustainable Chem. Eng.* **7**, 11050–11055 (2019).

[10] Kundra, M. et al. Continuous flow semi-hydrogenation of alkynes using 3D printed catalytic static mixers. *Chem. Eng. Process.* **154**, 108018 (2020).

10) Page 17, line 22: “weightlessness peak” – What’s a weightlessness peak, can the authors explain?

Reply: The weightlessness peak means the mass loss peak. The spent Pd/ZnO-400 catalyst has been detected by TG-DTG curve. As shown in Figure S8, there is no obvious mass loss peak in the high temperature region.

Modification: “weightlessness peak” has been replaced by “obvious mass loss peak” in the revised manuscript.

Additionally, it was noteworthy that no obvious mass loss peak was observed in the high-temperature region for the TG/DTG curve of spent Pd/ZnO-400 catalyst (as shown in Figure S13).

11) Page 20, section named ”In situ DRIFT spectroscopy of the hydrogenation of propargyl alcohol” – this section is very lengthy and parts of it should be moved to the supporting information.

Reply: Thank you very much for your comments. For shortening the section “In situ DRIFT”, some sentences have been deleted.

Modification: The sentences “since the adsorption characteristics of propargyl alcohol on the catalyst surface are considered to be an important factor in determining its selective hydrogenation.” , “Comparing the results from Figure 4(b) and Figure S16, this side reaction prefers to carry out over Pd/ZnO-150 sample.” have been deleted in the revised manuscript.

12) Page 28, line 8: “...and H₂-to-liquid feed ratio of 600.” – Is this feed ratio based on volume/volume or mass/mass?

Reply: The feed ratio is based on volume/volume in the manuscript.

Modification: The effects of reaction temperatures (RT-100 °C), H₂ pressures (0.1-2.0

MPa), the volume ratio of H₂-to-liquid feed (50-600), and the contact time (0-87 g_{cat.}·h·mol⁻¹) on the catalytic performances in the selectivity hydrogenation of BYD over Pd/ZnO-400 catalyst were explored.

Reviewers' comments:

Reviewer #1 (Remarks to the Author):

After the modifications, this manuscript might be accepted for the publication.

Reviewer #2 (Remarks to the Author):

In my opinion, a more rigorous and detailed scientific analysis was proposed. In its revised form the manuscript was acceptable. The authors answered exhaustively and in a satisfactory way all my questions, so the article can be considered suitable for the publication.

Reviewer #3 (Remarks to the Author):

The authors have adequately addressed all questions and points raised during the initial review. The changes made to the manuscript have improved clarity and added valuable information. I can recommend for the revised manuscript to be accepted for publication in CommsChem.